# On the capability of the future ALTIUS UV-VIS-NIR limb sounder to constrain modelled stratospheric ozone

Quentin Errera[1], Emmanuel Dekemper[1], Noel Baker[1], Jonas Debosscher[1], Philippe Demoulin[1], Nina Mateshvili[1], Didier Pieroux[1], Filip Vanhellemont[1], and Didier Fussen[1]

[1]Royal Belgian Institute for Space Aeronomy (BIRA-IASB), Brussels, Belgium

**Correspondence:** Quentin Errera (quentin.errera@aeronomie.be)

**Abstract.** ALTIUS (Atmospheric Limb Tracker for the Investigation of the Upcoming Stratosphere) is the upcoming stratospheric ozone monitoring limb sounder from ESA's Earth Watch programme. Measuring in the ultraviolet-visible-near infrared (UV-VIS-NIR) spectral regions, ALTIUS will retrieve vertical profiles of ozone, aerosol extinction coefficients, nitrogen dioxide and other trace gases from the upper troposphere to the mesosphere. In order to maximize the geographical coverage, the instrument will observe limb scattered solar light during daytime (i.e. bright limb observations), solar occultations at the terminator and stellar/lunar/planetary occultations during nighttime. This paper evaluates the constraint of ALTIUS ozone profiles on modelled stratospheric ozone by means of an Observing System Simulation Experiment (OSSE). In this effort, a reference atmosphere has been built and used to generate ALTIUS ozone profiles, along with an instrument simulator. These profiles are then assimilated to provide ozone analyses. A good agreement is found between the analyses and the reference atmosphere in the stratosphere and in the extra-tropical upper troposphere. In the tropical upper troposphere, although providing a significant information in the analyses, the assimilation of ozone profiles does not completely eliminate the bias with respect to the reference atmosphere. The impact of the different modes of observations have also been evaluated, showing that all of them are necessary to constrain ozone during polar winters where solar/stellar occultations are the most important during the polar night and bright limb data are the most important during the development of the ozone hole in the polar spring.

## 1   Introduction

Stratospheric ozone ($O_3$) is an essential component of the Earth's system. By absorbing solar ultraviolet (UV) light in the stratosphere, it protects the Earth surface from exposure to harmful radiation (Brasseur and Solomon, 2005). Surface emissions of halogen compounds, whose production has been progressively banned after the implementation of the Montreal protocol in 1987, are responsible for the reduction of the ozone layer worldwide (WMO, 2018). Emissions of long-lived greenhouse gases may also have direct – via the emissions of methane and nitrous oxide – and indirect – via change in atmospheric temperature – effects on the state of the ozone layer (SPARC/IO3C/GAW, 2019). Moreover, by affecting the thermal structure of the atmosphere, changes in the stratospheric ozone will have an impact on the atmospheric circulation (Hardiman et al., 2014). The monitoring of the stratospheric composition is thus crucial.

In the past 40 years, there have been a number of limb-viewing satellite instruments dedicated to stratospheric observations (SPARC, 2017). They have provided high-resolution vertical profiles of ozone and other key parameters (temperature, aerosols extinction, halogens, water vapor, ...) allowing to understand the causes affecting the ozone layer and their consequences. Today, only a few instruments are confirmed for a future launch. A rare example is the Ozone Mapping and Profiler Suite Limb Profiler (OMPS-LP, Flynn et al., 2006) onboard the Joint Polar Satellite System platform 2 (JPSS-2), scheduled for 2022, which will measure ozone profiles from bright limb observations.

Another instrument, to be launched in 2024, is the Atmospheric Limb Tracker for the Investigation of the Upcoming Stratosphere (ALTIUS, Fussen et al., 2019). This mission was proposed by Belgium as an element of ESA's Earth Watch programme, and was later supported by Canada, Luxembourg, and Romania. ALTIUS objectives are to observe the global distribution of stratospheric ozone, aerosol extinction, nitrogen dioxide and other trace gases at high vertical resolution. In addition, ALTIUS will also be a demonstration mission to deliver near real time (NRT) ozone profiles, i.e. with a latency of less than 3 hours from the sensing to the delivery of the retrieval product to operational services like the Copernicus Atmosphere Monitoring Service (CAMS, Lefever et al., 2015; Flemming et al., 2017).

Since its inception, ALTIUS was designed as an innovative hyperspectral imager to be flown onboard a micro-satellite of the PROBA class (Vrancken et al., 2014) operating from a sun-synchronous near polar orbit. The original trade-off between cost and scientific return led to the selection of a passive remote sensing instrument sensitive to the ultraviolet (UV), visible (VIS), and near-infrared (NIR) parts of the electromagnetic spectrum (Fussen et al., 2019). In order to maximize the spatial coverage of the geophysical products, three observation geometries will be applied to every orbit: bright limb measurements during daytime, solar occultations at the terminator and stellar/lunar/planetary occultations during night time.

The aim of this paper is to evaluate to what extent ALTIUS ozone profiles are able to constrain modelled ozone in a data assimilation system and to evaluate the added value of the different modes of observations. This is done using an Observations System Simulation Experiment (OSSE). OSSEs are typically designed to investigate the potential impacts of prospective observing systems using data assimilation techniques (Masutani et al., 2010). In OSSEs, simulated rather than real observations are used as the input of the data assimilation system. While OSSEs were usually applied to evaluate satellite instruments dedicated to meteorological use, several experiments also focused on satellite instruments measuring tropospheric chemical composition (Claeyman et al., 2011; Timmermans et al., 2015; Abida et al., 2017). To the best of our knowledge, this paper is the first discussing an OSSE for a satellite instrument dedicated to stratospheric ozone profile measurements.

In this paper, we set up several OSSEs to answer two questions related to ALTIUS observations. First, we would like to compare the constraint of assimilated ALTIUS ozone profiles on modelled ozone fields in the stratosphere and the upper troposphere against the constraint of ozone profiles measured by the Microwave Limb Sounder (MLS, Waters et al., 2006). MLS measures day and night $O_3$ (and other trace gases) with high accuracy and stability since 2004 (Hubert et al., 2016) and has been successfully assimilated by numerous chemical data assimilation system (e.g., Wargan et al., 2017; Inness et al., 2019; Errera et al., 2019). Second, we also want to measure the added value of the different ALTIUS modes of observation (solar, stellar, planetary and lunar occultations, and bright limb measurements) in particular for the stellar occultations which are the most complex to implement in the mission scenario.

**Table 1.** ALTIUS primary scientific objectives requirements for $O_3$ profiles. For the product uncertainty, the least stringent value of the relative or absolute uncertainty specifications shall apply. In the second column, "target" denotes the desired performance and "threshold" denotes the minimum satisfactory performance.

| Altitude range | Target/Threshold uncertainty | Coverage | L2 product latency | Applicability |
|---|---|---|---|---|
| 15-45 km | 5/20 % or 50/100 ppbv | all latitudes | <3 hours | NRT product |
| 20-45 km | 3/10 % or 50/100 ppbv | all latitudes | 4 weeks | Climatology-grade product |
| 15-45 km | 10/30 % or 50/100 ppbv | polar | 4 weeks | $O_3$ hole conditions |

To answer the first question, we have simulated five months of ALTIUS observations using a chemical data assimilation (DA) system dedicated to stratospheric ozone. In a first DA experiment, called the nature run, MLS $O_3$ profiles were assimilated and the analyses were saved in the simulated ALTIUS space of observations. These simulated ALTIUS observations were then perturbed according to the estimated ALTIUS ozone error covariance matrices. Simulated ALTIUS data were then assimilated (the assimilation run). The impact of ALTIUS ozone profiles was evaluated by measuring how the assimilation run could reproduce the nature run. A control run, without data assimilation, was also performed to check that the agreement between the nature and assimilation runs is due to the constraint of ALTIUS profiles and not due to the model alone.

To answer the second question, several assimilation experiments have been carried out using only one or two of the measurement modes of ALTIUS. Comparison of these experiments with the assimilation run emphasizes the importance of each observation mode.

This paper is organized as follows: Section 2 provides additional information on the ALTIUS mission. Section 3 introduces the DA system and its configurations for this study. In Sect. 4, the setup of the OSSE is described i.e. the nature run, the control run, the simulation of ALTIUS profiles, the assimilation run and the runs with the selected ALTIUS modes of observation. The evaluation of the assimilation run is presented in Sect. 5, while Sect. 6 evaluates the impact of the different ALTIUS observation modes. Conclusions are provided in Sect. 7.

## 2 The ALTIUS Mission

As mentioned in the introduction, one of the objectives of ALTIUS is the measurement of vertical ozone profiles by means of hyperspectral imagers operating in the UV, VIS and NIR with three measurement geometries:

1. On the day side of the orbit, limb-scattered solar light will be measured by looking above the Earth's horizon at tangent altitudes ranging from 0 to 100 km. This method was successfully applied by previous missions such as OSIRIS (Llewellyn et al., 2004), SCIAMACHY (Bovensmann et al., 1999), SAGE-III (Rault, 2005), and OMPS-LP (Flynn et al., 2006), where only instruments measuring in UV, VIS and NIR are mentioned in this list.

2. Close to the terminator of the orbit, the instrument will point at the Sun and track its occultation across the atmosphere (one sunset and one sunrise per orbit). This method was applied by the family of SAGE instruments (Mauldin-III et al., 1985) and SCIAMACHY. Given the sun-synchronous polar orbit of ALTIUS, these measurements will return $O_3$ profiles only at high latitudes.

3. On the night side of the orbit, the instrument will point at stars, planets or the Moon, and measure their apparent ascent/descent through the atmosphere. Only applied by GOMOS (Kyrölä et al., 2004) for stars and planets or SAGE-III for the Moon (Chu et al., 2002), this method will allow ALTIUS to return several profiles on the night side of the orbit, which are usually left unmeasured by other UV-VIS-NIR sensors.

The scientific requirements for ozone observations are summarized in Table 1. The vertical resolution of ALTIUS ozone profiles will be around 1-2 km for solar occultations and 2-3 km for the other observation modes. Further details on the instrumental concept can be found in Montrone et al. (2019) and Fussen et al. (2019).

Compared to all previous UV-VIS-NIR limb sounders (limb-scatter as well as occultation), the ALTIUS mission concept has two important assets. First, the native imaging capability alleviates the need for complex, heavy, and failure-prone limb scanning or Sun/star tracking mechanisms. Tangent altitude registration calibration, which is a frequent cause of severe biases in the retrieved $O_3$ profiles (e.g. Moy et al., 2017), is also made simpler as the entire scene is captured in every acquisition. Second, the capability of ALTIUS to perform $O_3$ measurements in the three observation geometries will enable the densest sampling ever achieved by a UV-VIS-NIR sounder. It also enables self-validation by comparing almost collocated limb and solar occultation observations, for instance.

## 3  The BASCOE Data Assimilation System

In this study, model simulations and assimilation runs have been done using the Belgian Assimilation System for Chemical ObsErvations (BASCOE, Errera and Fonteyn, 2001; Errera et al., 2008, 2019). This system is based on a chemistry transport model (CTM) dedicated to stratospheric composition. While past publications were based on a CTM including 60 chemical species interacting via around 200 chemical reactions, the present study uses the COPCAT linearized ozone chemical scheme (Coefficients for Ozone Parameterization from a Chemistry And Transport model, Monge-Sanz et al., 2011) in order to speed up the computing time. The COPCAT scheme uses a simple linear expression to relax ozone towards an ozone climatology, this climatology being calculated using a 3-dimensional CTM with full stratospheric chemistry. Compared with other linearized schemes, COPCAT chemistry has the advantage of providing a better representation of polar ozone depletion and displays good agreement with ozone observations (Monge-Sanz et al., 2011; Jeong et al., 2016). Nevertheless, like many other linear schemes, COPCAT underestimates middle stratospheric ozone (Monge-Sanz et al., 2011) because of biases in its ozone climatology. More recently, it was also pointed out that COPCAT chemistry underestimates tropospheric ozone (Dragani et al., 2018), which will be discussed in Sect. 5.

Wind and temperature fields used to drive the system come from the ERA-Interim reanalysis (Dee et al., 2011). The model time step is set to 30 minutes and the spatial resolution depends on the numerical experiments performed for this paper (see Sect. 4 and Table 2).

The observation operator of BASCOE consists of a linear interpolation of the model state to the geolocation of the observed profile tangent points available at the model time $\pm$ 15 minutes (i.e. half of the model time step). It has been used to save the BASCOE state in the space of all observations used in this study (i.e. ozonesondes and satellite data) including the simulated space of ALTIUS observations (described in Sect. 4.3.1). Averaging kernels have not been applied in this study since the BASCOE Ensemble Kalman Filter (EnKF) is not ready for their use. The vertical resolution of these observations is sufficiently high – and similar to the model vertical resolution – that their use is typically considered unnecessary for ozone profiles (Errera et al., 2019).

Two data assimilation methods are available in BASCOE, the four-dimensional variational method (4D-Var, Errera and Ménard, 2012) and the EnKF (Skachko et al., 2014, 2016), the latter one being used here. The BASCOE EnKF setup used in this paper is similar to the one used in Errera et al. (2019). The system considers an ensemble of 20 model states initialized by adding 20% error perturbations to the initial conditions. During the assimilation, each member is perturbed by adding 2.5% noise at every model time step. When available, observations are assimilated at every model time step. The observational errors are scaled using a profile estimated by the Desroziers method (Desroziers et al., 2005), allowing to have $\langle \chi_k^2/m_k \rangle \sim 1$ where $\chi_k^2$ measures the difference between the assimilated observations and the model forecasts weighted by their combined error covariances; $m_k$ is the number of observations at time step $k$ and $\langle . \rangle$ denotes the mathematical expectation (as in Skachko et al., 2014; Errera et al., 2019). The observational error scaling factor calculated for the nature and assimilation runs are shown in Fig. 11 and will be discussed in Sect. 5.

## 4 The OSSE setup

OSSEs are setup using at least three numerical experiments (Masutani et al., 2010; Timmermans et al., 2015): the nature run (NR), the control run (CR) and the assimilation run (AR). The nature run, along with an instrument simulator, defines the true state of the atmosphere from which observations of the new instrument are simulated. The control run, being either a free model run or a data assimilation run without (all or a part of) the simulated observations, provides the baseline atmospheric state and will be used to evaluate the added value of the new instrument. The assimilation run considers the new instrument in addition to the other assimilated data used in CR, if any. To avoid the identical twin problem that can lead to overoptimistic results, the assimilation and control runs should use a different model than the nature run. This ensures that the similarity between the NR and AR trajectories is most likely due to the assimilation of the new instruments.

In this study, the set-up is slightly different. All runs use the same BASCOE model (see Sect. 3) and the identical twin problem is solved as follows. First, the nature run is calculated using a higher spatial model resolution than the control and assimilation runs. Also, the nature run is based on the assimilation of MLS $O_3$ observations using BASCOE EnKF which are not assimilated in the control and assimilation runs. The control run is based on a BASCOE model simulation (no assimilation)

**Table 2.** List and configuration of BASCOE numerical experiments used in this study.

| Label | Period (in 2009) | Init. Cond. | Resolution[1] | Observations |
|---|---|---|---|---|
| NR | 15 May-26 Oct. | BRAM2 | 144x91x60 | MLS |
| CR | 15 May-26 Oct. | BRAM2 | 96x73x60 | None |
| AR | 1 Jun-26 Oct. | CR | 96x73x60 | ALTIUS all modes |
| LIMB | 1 Jun-26 Oct. | CR | 96x73x60 | ALTIUS limb |
| LSo | 1 Jun-26 Oct. | CR | 96x73x60 | ALTIUS limb + solar occultations |
| LSt | 1 Jun-26 Oct. | CR | 96x73x60 | ALTIUS limb + stellar occultations |
| SoSt | 1 Jun-26 Oct. | CR | 96x73x60 | ALTIUS solar + stellar occultations |
| $MLS_{All}$ | 1 Jun-26 Oct. | CR | 96x73x60 | MLS |
| $MLS_{Day}$ | 1 Jun-26 Oct. | CR | 96x73x60 | MLS daytime (i.e. sza<85°) |

[1] i.e. $N_{longitude} \times N_{latitude} \times N_{level}$. Thus 144x91x60 corresponds to 2.5° longitude×2° latitude×60 levels and 96x73x60 corresponds to 3.75° longitude×2.5° latitude×60 levels.

using the same initial conditions as NR, on the same day. The assimilation run is based on the assimilation of the simulated ALTIUS $O_3$ observations also using BASCOE EnKF. It is running with the same spatial resolution as CR and is initialized 15 days later than NR and CR with the $O_3$ state from CR as initial conditions (the 15-day delay allows to have initial conditions sufficiently departed from NR initial conditions). This ensures that the similarities between NR and AR are only due to the assimilation of the simulated ALTIUS observations in AR. While OSSEs will in general measure the value of a new instrument added in an existing observing system, our goal is more to measure how the new instrument (i.e. ALTIUS) could replace an old one (i.e. MLS). This is why the control and the assimilation runs do not assimilate MLS data.

The setup of these experiments is summarized in Table 2 and detailed in the following subsections, as well as the simulation of ALTIUS observations. Several additional experiments are also summarized in Table 2 and detailed at the end of this section.

## 4.1 The nature run

The nature run (NR) is based on the BASCOE EnKF assimilation of observations taken by the Microwave Limb Sounder (MLS, Waters et al., 2006) operating on NASA's Aura satellite. MLS measures vertical profiles of around fifteen chemical species, including ozone, during day and night. Here, we have used MLS data version 4.2 where observations are accepted/rejected following the guidelines given in Livesey et al. (2020). Ozone profiles have a vertical range of validity between 0.02 and 261 hPa such that results below that level will not be discussed. MLS individual profile precision error, in percent, is given in Fig. 7e where ranges of values are given below 68 hPa. In the middle stratosphere (∼1 to 68 hPa), the precision is generally below 5%. It increases in the upper troposphere lower stratosphere (UTLS) to values between typically 5 to 100%. In volume mixing ratio (vmr) units, this large range corresponds to values between 0.02 and 0.03 ppmv such that a high percentage error corresponds generally to ozone in the tropical tropopause layer (TTL) where the ozone abundance can be as low as the MLS precision.

The nature run has a resolution of 2.5° in longitude by 2° in latitude by the 60 levels of ERA-Interim (i.e. the resolution is 144x91x60 grid points). It is initialized on May 15, 2009 at 0 UT with the ozone state from the BASCOE Reanalysis of Aura

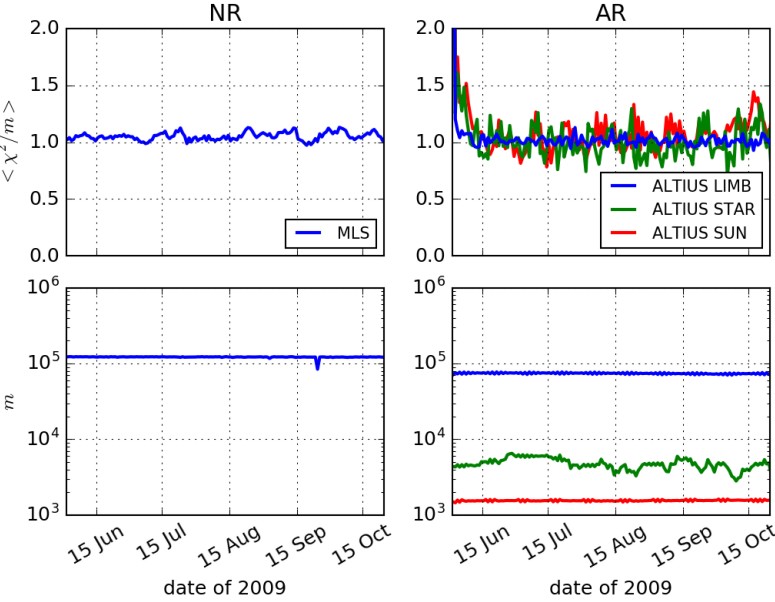

**Figure 1.** Time series of the $\chi^2$-test (i.e. the daily mean $\chi^2$ divided by the number of assimilated observations $m$, top row) and the number of assimilated observations (bottom row) for the nature run (NR, left column) and the assimilation run (AR, right column).

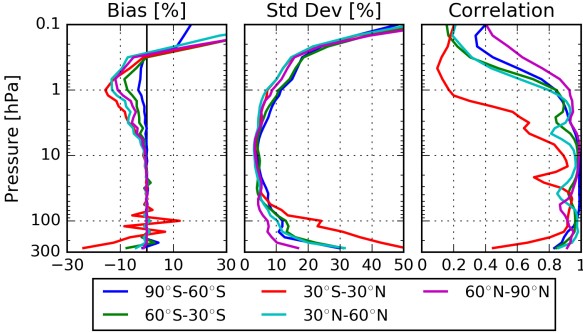

**Figure 2.** Forecast minus observation (FmO) statistic profiles between the nature run (NR) and the assimilated MLS data in five latitude bands (see legend) for the period June-October 2009. Left: bias (or mean difference) between NR and MLS. Center: the standard deviation of the differences. Right: the sample correlation coefficients between NR and MLS. Differences are shown in % and are normalized by the mean of MLS.

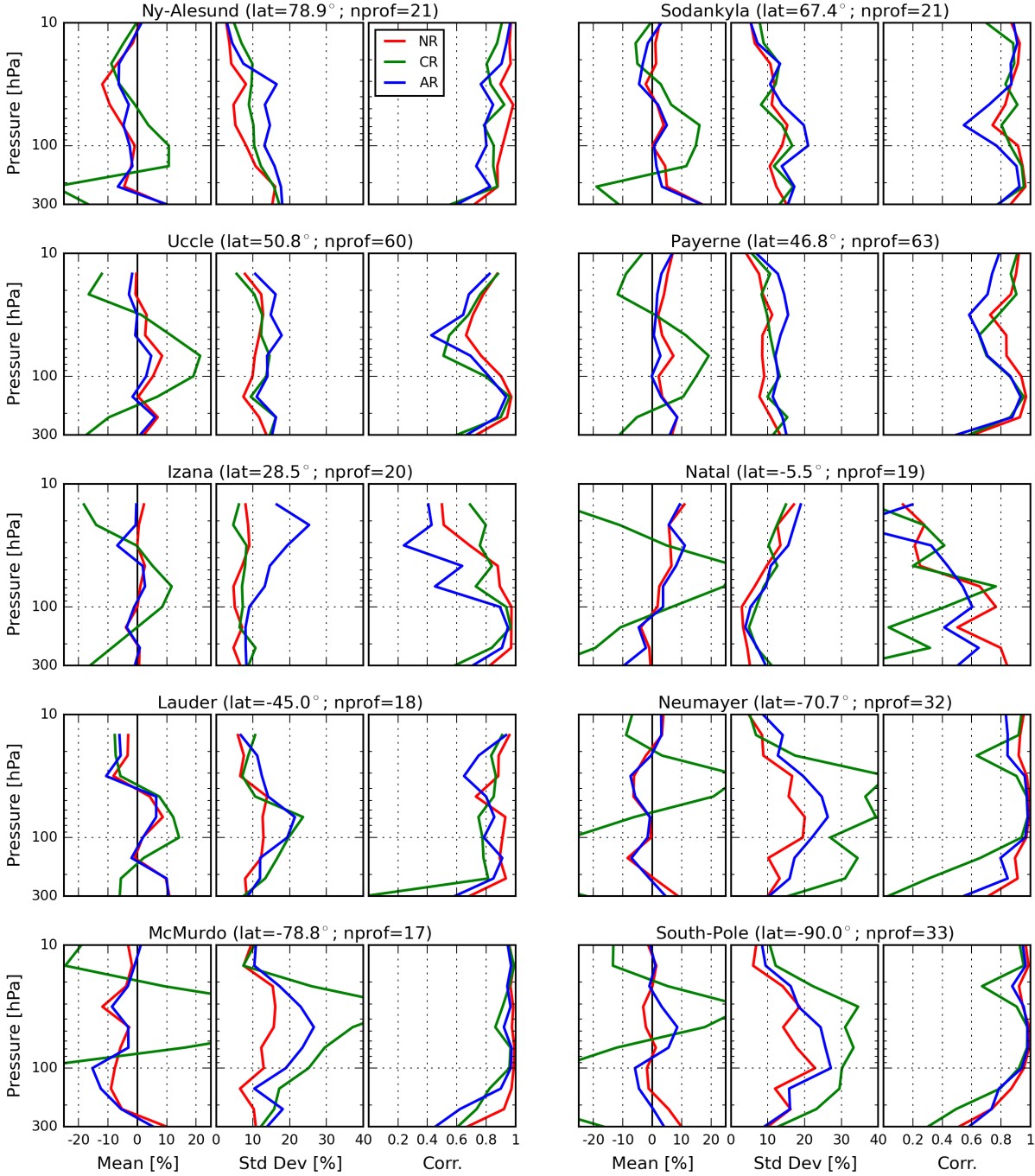

**Figure 3.** Comparison between ozonesonde profiles and the nature run (red line), the control run (green line) and the assimilation run (blue line) for the period $1^{st}$ June-$25^{th}$ October 2009 at 10 stations of the Network Detection for Atmospheric Composition Change (NDACC). For each station are shown: the mean differences between the BASCOE runs and the ozonesondes normalized by the ozonesonde values (in %, left plot), the associated standard deviation (in %, centre plot) and the sample correlation coefficients between the BASCOE runs and ozonesondes (right plot). The latitude of the station and the number of soundings is given in the title of each group of three plots.

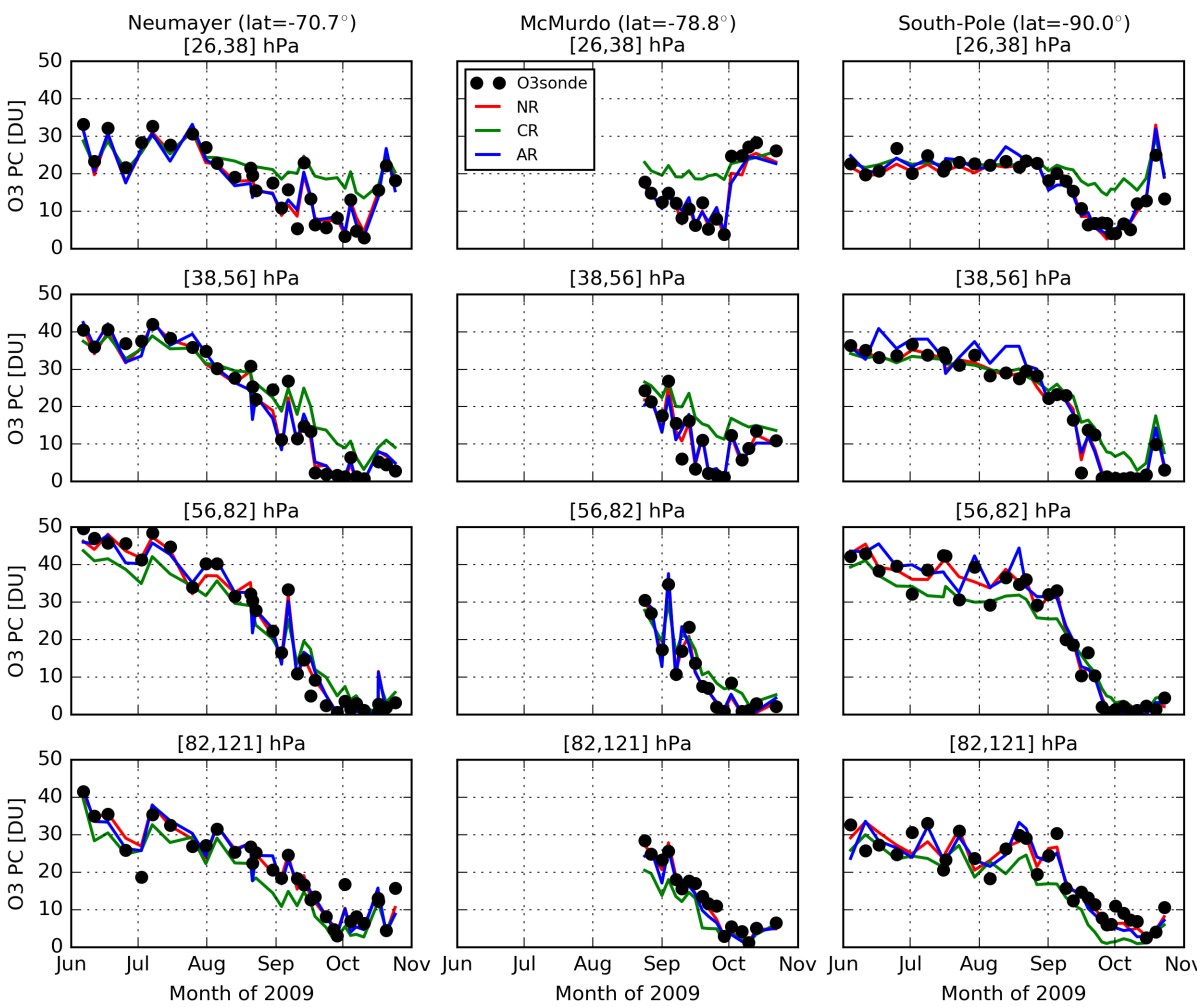

**Figure 4.** Time series of ozone partial column between June-October 2009 at four different pressure layers in the lower stratosphere and at three Antarctic NDACC stations from ozonesondes observations (black line), the nature run (NR, red line), the control run (CR, green line) and the assimilation run (AR, blue line).

MLS, version 2 (BRAM2, Errera et al., 2019). The assimilation ends on October 26, 2009 at 0 UT, because MLS ozone on October 26 and 27 are not suited for scientific studies (see also Fig. 6). During the production of the nature run, the model ozone state is saved at the geolocation of ALTIUS observations, as described below.

The success of MLS assimilation is verified by means of the $\chi^2$-test and the forecast minus observation (FmO) statistics (results of CR and NR shown in the following figures will be discussed in later sections). The time series of the daily mean $\chi^2$-values (discussed in Sect. 3) of NR are close to one (see Fig. 1) which confirms the internal consistency of the BASCOE EnKF setup for the assimilation of Aura MLS. The FmO statistics against MLS tell us how BASCOE is able to forecast the assimilated data prior assimilation (see Fig. 2). The agreement is generally very good, with biases lower than 10%, standard deviations lower than 20% and correlations higher than 0.8. There are, however, several exceptions. Above (i.e. at pressures lower than) 1 hPa, the statistics can be worse due to bias in the COPCAT ozone chemistry (which is also the case in many other models, see discussion in Errera et al., 2019). For this reason, AR results will not be discussed above 1 hPa. In the TTL (i.e. at pressures higher than 50 hPa) where MLS ozone has a relatively large error, statistics are also not as good. The bias profile in the TTL also displays vertical oscillations which are due to oscillations in the MLS profiles. BASCOE vertical resolution being lower than that of MLS ozone profiles, the system finds a compromise which eliminates these oscillations (this issue is also discussed in Errera et al., 2019). Results in the TTL will be discussed in more detailcd in Sect. 5.

NR is also evaluated against independent observations from ozonesondes (Figs. 3 and 4) whose uncertainties are assumed random and uncorrelated, and around 5% in the stratosphere, 7-25% around the tropopause and 5-10% in the troposphere (Sterling et al., 2018). In most cases, the nature run and ozonesonde observations agree within ±10% with a correlation better than 0.8, which is good. The nature run clearly captures the ozone depletion that occurs above Antarctica as observed by ozonesondes (Fig. 4). The nature run also agrees well with independent satellite observations by the Atmospheric Chemistry Experiment Fourier Transform Spectrometer v3.6 (ACE-FTS, Bernath et al., 2005; Boone et al., 2013, see Fig. S1 in the supplement) and the Michelson Interferometer for Passive Atmospheric Sounding v220 (MIPAS, Fischer et al., 2008; von Clarmann et al., 2009, see Fig. S2). The good agreement between NR and independent observations validates the nature run and justifies its use to simulate ALTIUS observations.

## 4.2 The control run

The control run (CR) is based on a BASCOE free model simulation (no assimilation) with a lower horizontal resolution than in NR: 3.75° in longitude by 2.5° in latitude by 60 levels of ERA-Interim (i.e. the resolution is 96x73x60 grid points). It starts on the same date as NR using the same initial conditions from BRAM2. Compared with ozonesondes, ACE-FTS and MIPAS, CR displays larger differences than NR, highlighting the added value of the assimilation of MLS in the nature run (see Figs. 3, 4, S1 and S2). However, note the relatively good representation of Antarctic ozone depletion in CR, thanks to the COPCAT chemistry, when compared to ozonesondes (see Fig. 4).

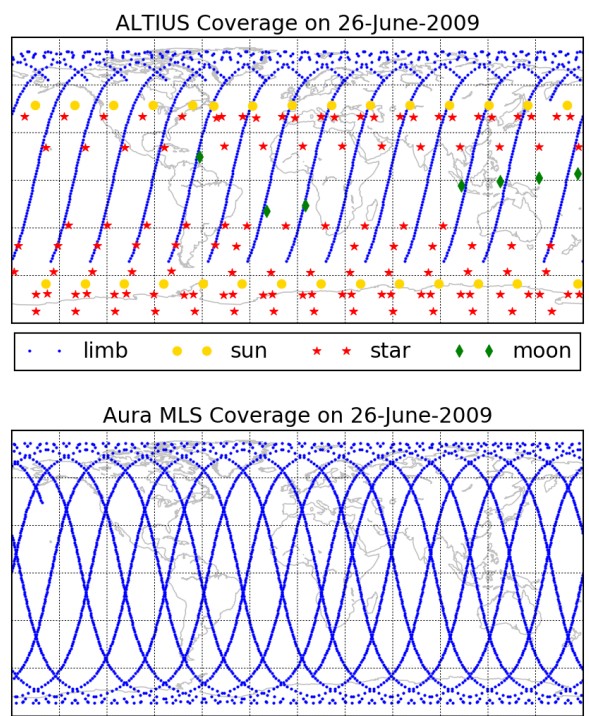

**Figure 5.** Simulated daily spatial coverage of the four ALTIUS modes (top) as compared with Aura MLS (bottom) on June 26, 2009.

## 4.3 Simulating ALTIUS observations

### 4.3.1 Profile geolocation

The geolocation (position and timing) of the ALTIUS measurements have been simulated with a circular sun-synchronous orbit at an altitude of 700 km and with a mean local time of the ascending node crossing at 10 PM (inclination being 98.19° and the initial period being 98.6 minutes). This corresponds to a revisit time of 17 days. The orbit was propagated for one year with a numerical propagator based on the Orekit space flight dynamics library (Maisonobe et al., 2018).

On the night side of the orbit, ALTIUS observes the rises and sets of the stars, planets or the Moon through the Earth atmosphere. All stars brighter than a visual magnitude of 1.5 (total of 23 stars) were selected for this simulation, in addition to the planets Mercury, Venus, Mars, Jupiter and Saturn, and the Moon, i.e. a total of 29 targets. The stellar positions as well as other characteristics were extracted from the ESA HIPPARCOS star catalogue (ESA, 1997). The stellar effective temperatures were taken from the SIMBAB astronomical database (Wenger, M. et al., 2000). The planets, the Moon and the Sun positions were computed with the NASA-JPL SPICE toolkit and their associated kernels (Acton, 1996; Acton et al., 2018). Note that planetary and stellar occultations will be measured and retrieved in the same way such that in the following part of the paper, both types of occultations will simply be referred to as stellar occultations.

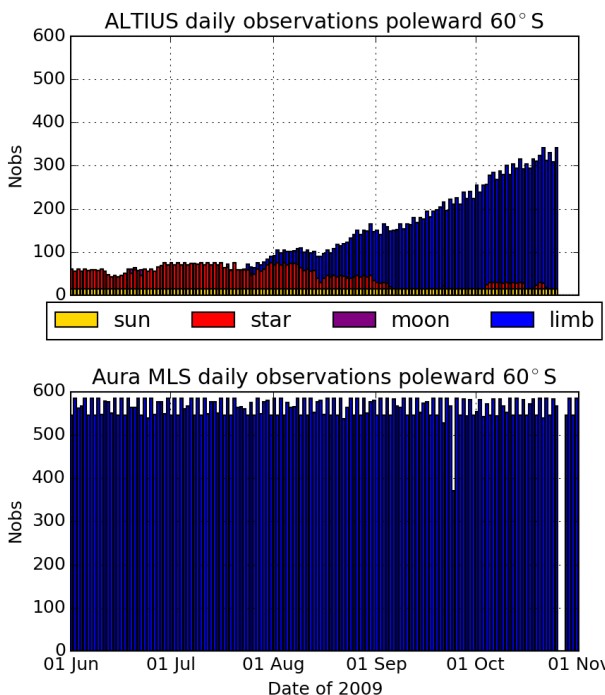

**Figure 6.** Number of simulated ALTIUS daily profiles for the four observing modes (top) as compared with Aura MLS (bottom) for the period June-October 2009.

The timing of the ALTIUS measurements is as follows: during daytime, for solar zenith angle (sza) smaller than $85°$, the instrument takes a set of spectral images of the bright limb every 30 seconds, in backwards-looking mode which provides a profile every $\sim$200 km. When the Sun line-of-sight (LOS) reaches a tangent altitude of 100 km, ALTIUS makes observations of the sunset, until the Sun has completely set (refracted altitude of 0 km). Then, observations of the rises and sets of the stars, planets and Moon begin. These measurements also occur between 0 to 100 km tangent altitudes of the lines of sight. The next target to observe will simply be the closest, in angular distance, to the current spacecraft LOS. The simulation is constrained by the maximum angular speed of one degree per second and a spacecraft stabilization time of 30 seconds. Nighttime observations stop just before sunrise, and solar occultation is performed up to 100km tangent altitude. Finally, bright limb measurements resume (when sza<$85°$). This procedure provides the geolocation of latitude, longitude and time of ALTIUS simulated profiles at the 30 km altitude tangent point. Variation of the latitude/longitude/time with the altitude of the tangent point has not been taken into account. At each ALTIUS geolocation, a profile from 0 to 100 km with a step of 1 km is considered. The ALTIUS space is thus defined by the one-dimensional vertical grid and the one-dimensional time/latitude/longitude vectors.

Figure 5 highlights the typical daily coverage of ALTIUS in June (during the polar night) and compares it with MLS. On that day, ALTIUS provides around 1300 bright limb profiles, 30 solar occultations, 120 stellar occultations and 7 lunar occultations which is less than half the number of MLS profiles for that day ($\sim$3500). At latitudes poleward of $60°$S, ALTIUS

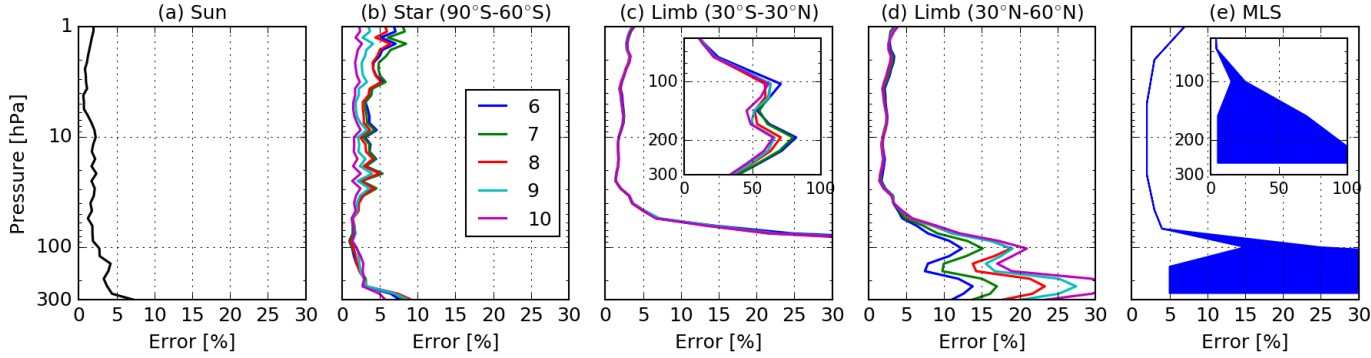

**Figure 7.** Typical simulated ALTIUS ozone error standard deviation profiles (in %) for solar (a) and stellar occultations (b), and bright limb (c and d). The color code indicates the month number. Stellar occultation error standard deviation profiles correspond to the 90°S-60°S bin. For bright limb, error profiles correspond to the 30°S-30°N (c) and 30°N-60°N (d) latitude bins, both for the same 30°-60° sza bin. For comparison, MLS single profile precision error standard deviations taken from Livesey et al. (2020) are shown in panel (e) with a range of min/max values below 68 hPa. A zoom in the tropical tropopause layer is shown in panels (c) and (e).

observations are only provided by the solar and stellar occultation modes during the polar night, reaching 15 and 50 daily profiles, respectively, while MLS provides around 550 daily profiles (see Fig. 6). Limb observations in the South Pole regions start again around the end of July.

### 4.3.2 Ozone profiles

Ideally, ALTIUS ozone profiles should be calculated as follows. Given the nature run saved in the space of ALTIUS, a radiative transfer model would be used to simulate the radiometric field seen by ALTIUS at every observation location. An instrument model would then be used to compute the raw signal which forms the level-0 (L0) data. Then, the data would undergo radiometric correction and georeferencing to form the level-1 (L1) product. Finally, ozone profiles, which constitute the level-2 (L2) products, would be retrieved.

Such a complex simulation is, however, not possible because part of the instrument model is still under development, in particular for the simulation of the L1 measurement noise. To overcome this issue, the simulated measurement noise strictly follows the signal-to-noise ratio (SNR) requirements specified for each ALTIUS observation mode. These SNR tables, which are close to those from previous UV-VIS-NIR limb sounders (Bovensmann et al., 1999; Rault and Xu, 2011; Bourassa et al., 2012), are provided in the supplement.

An additional shortcoming is the unaffordable computation time to simulate the L0 measurements considering the ~1450 daily ALTIUS profiles (currently ~1 minute of computation time per profile for the bright limb mode). To overcome this issue, a sample of L2 error covariance matrices has been calculated for a number of ozone profile conditions representative of the OSSE period (June-October 2009). These matrices are then used to perturb the NR state saved in the ALTIUS space, thus providing the ALTIUS simulated profiles. The error covariance matrices are obtained by linear propagation of the L1 error

covariance matrix by use of the gain matrix of the inverse model (Rodgers, 2000).

The sample set of ozone profiles has been built for the following conditions. For stellar occultations, we have defined five latitude bins (90°S-60°S, 60°S-30°S, 30°S-30°N, 30°N-60°N and 60°N-90°N) and five-month bins (for the five months of the OSSE). An ozone profile is associated with each bin, which corresponds to the mean of the nature run saved in the ALTIUS stellar occultation space. A similar approach is followed for bright limb observations where four additional bins have been

added to take into account the variation of the solar zenith angle (0°-30°, 30°-60°, 60°-75°and 75°-85°). For solar occultations, only one error profile has been calculated from the mean of the nature run in the space of ALTIUS solar occultation in July and in the southern hemisphere. Note that for star and bright limb measurements, some bins are empty, e.g. for bright limb inside the South Pole bin during the polar night or for stars in the North Pole bin during the polar day.

Figure 7 shows typical standard deviation error profiles for solar occultations, stellar occultations in the South Pole bin,

bright limb in the tropical bin and the 30°N-60°N bins. For stars and bright limb, profiles for the five months are shown. All bright limb profiles correspond to the 30°-60°sza bin. Between 1 and 50 hPa, uncertainties are relatively small, below the targeted 5% shown in Table 1. Below 50 hPa, uncertainties increase especially in the Tropics for bright limb measurements. MLS profile precision errors taken from the MLS data quality document (Livesey et al., 2020) are also shown in Fig. 7. Having defined ALTIUS error covariance matrices for the different ALTIUS modes of observation and for different ozone conditions,

ALTIUS simulated L2 profiles are generated as follows:

1. For each profile of the nature run saved in the ALTIUS space, a Gaussian noise profile $n$ is calculated with zero mean and a standard deviation set to one.

2. This noise profile $n$ is multiplied by the square root of the error covariance matrix $\mathbf{S_z}$ of the corresponding ALTIUS mode and ozone condition: $n' = \mathbf{S_z}^{1/2} n$

3. The simulated profile $y'$ is calculated by adding $n'$ to the NR state $y$ saved in the ALTIUS space during the NR production: $y' = y + n'$

Note that the simulated ALTIUS profiles will inherit some of the noise of the NR experiments. This is due to the fact that NR is produced by an EnKF which adds a small perturbation to the ozone field before each analysis step (see Sect. 3). This has some implications on the results discussed in Sect. 5.

**4.4  The assimilation run**

The reference assimilation run (AR) is based on EnKF assimilation of ALTIUS simulated observations using BASCOE. It uses the same spatial resolution as CR but starts 15 days later (on June 1, 2009) and is initialized by the CR ozone state at that time. Figure 1 shows the time series of the $\chi^2$-test for the assimilation run (AR). For all observation modes, the values converge toward 1 while stellar and solar occultations display larger daily variability than the bright limb, likely due to the lower number

of observations for these modes. Note that due to their relatively low number, lunar occultations have not been assimilated in this study.

As done with NR and CR, AR has been compared with ozonesondes (Figs. 3-4), ACE-FTS (Fig. S1) and MIPAS (Fig. S2). On average, AR shows lower biases with respect to these independent observations than CR, very close to the agreement to

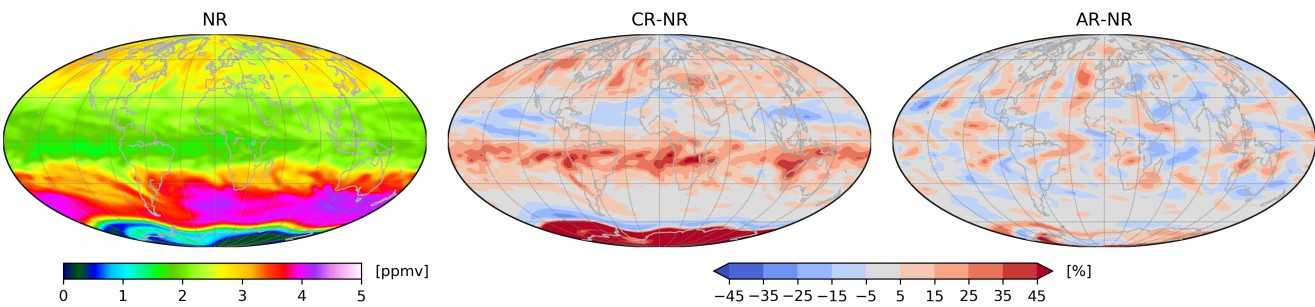

**Figure 8.** Left: Daily average ozone distribution (in ppmv) from the nature run (NR) around ~44 hPa (i.e. model level 21) on September 15, 2009. Center: the differences between the control run (CR) and the nature run normalized by the nature run (in %). Right: as center but for the assimilation run (AR).

NR, especially in the South Pole region. However, the standard deviations of the differences for AR are significantly larger than for NR and CR and with weaker correlations. While these comparisons tell us how AR could reproduce real observations, AR evaluation should be done with comparison against NR which represents the best approximation of the truth, this is done in Sect. 5.

## 4.5    Additional runs

Six additional BASCOE experiments have been carried out using the same model configuration as AR (see Table 2). Four of them are done in order to evaluate the impact of the different ALTIUS modes of observation in the assimilation run. These experiments assimilate ALTIUS bright limb mode (labelled LIMB), limb and solar occultations (LSo), limb and stellar occultations (LSt) and solar and stellar occultations (SoSt). The last two experiments have been run in order to evaluate the noise inherited by ALTIUS simulated data from NR and the impact of the limited sampling of ALTIUS during nighttime. These experiments consider (1) MLS all data (which differs from NR in the experiment configuration, see Table 2) and MLS daytime observations only (sza<85°).

## 5    Evaluation of the Assimilation Run

Figure 8 illustrates the agreements and differences between the nature, the control and the assimilation runs. It shows the ozone distribution for the nature run on September 15, 2009 at around 44 hPa (i.e. in the lower stratosphere) and the differences between the nature run and the control and assimilation runs. At that time, a large part of polar ozone has been destroyed by active chlorine above Antarctica, as shown by the low ozone abundances in the nature run in this region. We see that the biases in CR have been largely reduced in AR, in particular above Antarctica and in the Tropics.

Another illustration of the agreements and differences between NR and {CR,AR} is provided in Fig. 9, showing zonal mean ozone profiles in the South Pole region and in the Tropics on September 15, 2009. Above the South Pole, NR ozone shows a minimum around 40 hPa due to ozone destruction by active chlorine which is well captured by AR. This is not the case

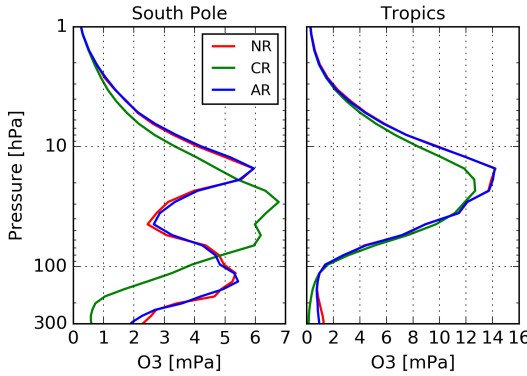

**Figure 9.** Daily zonal mean ozone profiles (in mPa) from the nature run (NR, red line), the control run (CR, green line) and the assimilation run (AR, blue line) above the South Pole (poleward of 70°S, left plot) and in the Tropics (between 20°S-20°N, right plot) on September 15, 2009.

with CR where ozone destruction seems to be small at that time (see the ozone minimum around 50 hPa). In the Tropics, CR underestimates NR ozone between 5 and 30 hPa, while AR and NR are in good agreement. Below (i.e. at pressures above) 100 hPa, CR also underestimates NR while AR displays a large reduction of this bias. The origin of the negative biases in CR are due to biases present in the linearized COPCAT chemical scheme as discussed in Sect. 3.

Figures 8-9 provide qualitative comparisons between NR, CR and AR. For quantitative comparisons, we use three statistical
indicators: the normalized mean biases (NMB, in %) between NR and {CR,AR}, the associated normalized standard deviations (NSD, in %) and the sample correlation coefficients between NR and {CR,AR}. NMB and NSD use the mean values of NR for normalization. Figure 10 compares NR against CR and AR using these three statistical indicators for the period June 15-October 25, 2009, i.e. starting 15 days after the initial date of AR to exclude the spin-up period.

CR underestimates NR at altitudes above (i.e. pressures below) 20 hPa and in the troposphere while it overestimates NR in
the lower stratosphere and in the South Pole region. Standard deviations are below 10% in the middle stratosphere and increase to more than 20% and 50% at the South Pole and in the tropical upper stratosphere, respectively. The correlations between NR and CR are good (>0.8) at mid- and high-latitudes and in the lower stratosphere.

The assimilation of simulated ALTIUS profiles improves most of these statistics when comparing AR with NR. Except in the tropical upper troposphere, biases are generally reduced to below ±5% with standard deviations lower than 15%. In the
tropical upper troposphere, biases in AR are largely reduced compared to CR, but remain significant with values that can be as high as 35%. In the same region, NSDs in (AR-NR) remain significant (∼30%) but show large improvements when compared to CR (∼50%). The correlations between AR and NR are also better than those between CR and NR below 100 hPa. In the middle stratosphere, the correlation remains weak between NR and both AR and CR which is discussed below. Finally, we have also checked that the improvement from CR to AR is statistically significant using the two-sample hypothesis z-test with
a confidence interval of 95% (not shown).

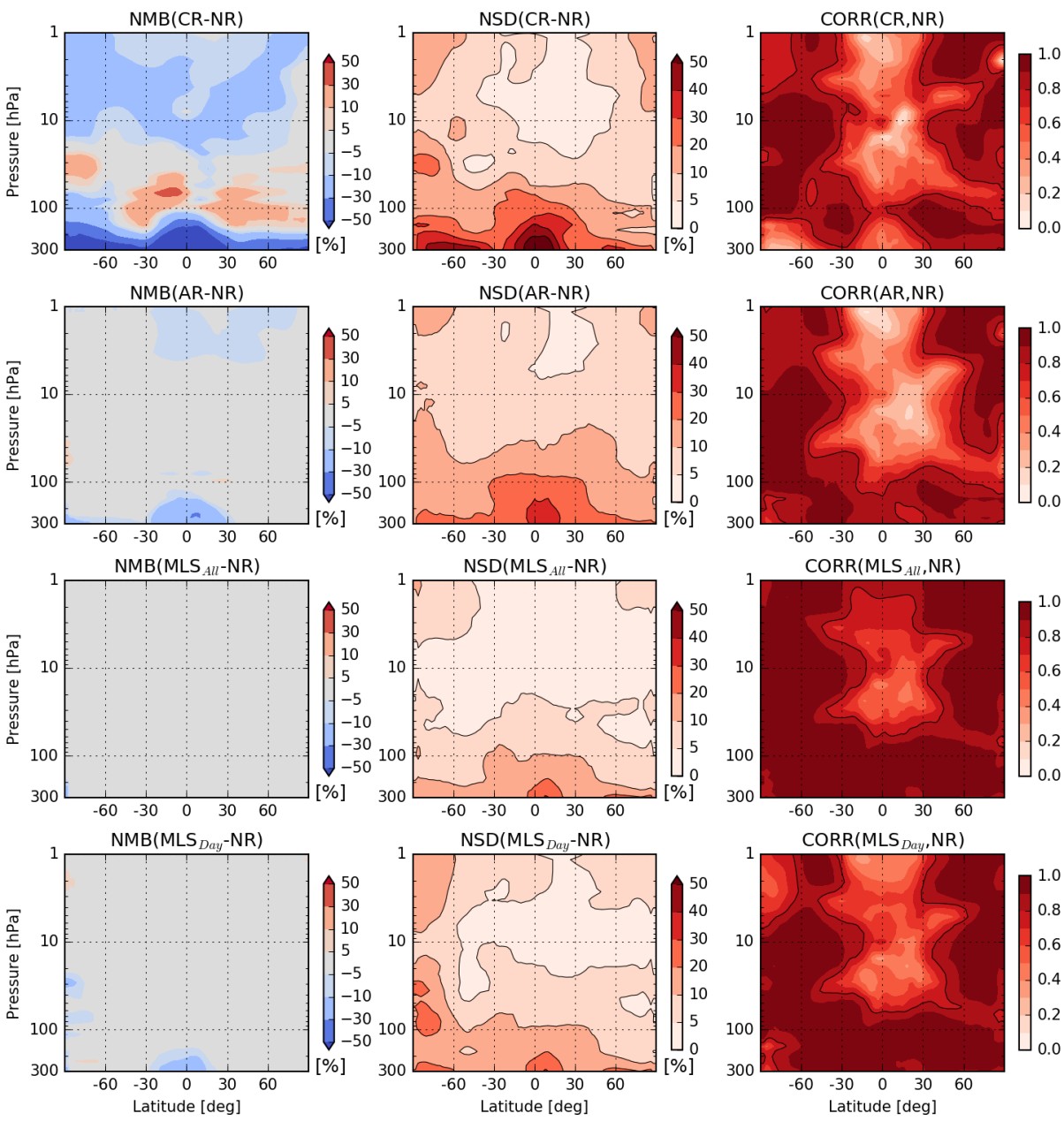

**Figure 10.** Top row: Statistical differences between NR and CR for the period June 15-October 25, 2009: the mean bias (NR-CR) normalized by the mean of NR (NMB, in %, left), the associated normalized standard deviation (NSD, in %, center) and the sample correlation coefficients between NR and CR (CORR, dimensionless, right). Black isolines the in center plots are shown between each % decade while in the right plot a single isoline shows the 0.8 correlation. Second row: The same as the top row, but for the comparison between NR and AR. Third row: The same as the top row, but for the comparison between NR and MLS$_{All}$. Fourth row: The same as the top row, but for the comparison between NR and MLS$_{Day}$.

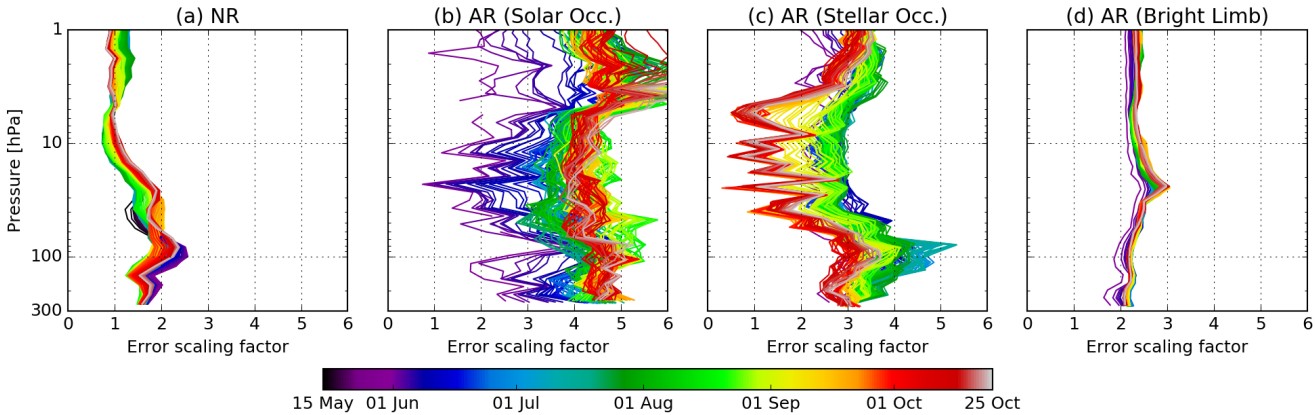

**Figure 11.** Profiles of daily error scaling factors estimated using the Desroziers method for MLS data assimilated in the nature run (NR, a), and ALTIUS data from solar occultations (b), stellar occultations (c) and bright limb (d) assimilated in the assimilation run (AR).

In order to infer the impact the EnKF noise and the low number of ALTIUS nighttime observations in the comparison between AR and NR, NR has been compared to two additional experiments: $MLS_{All}$ and $MLS_{Day}$ (described in Sect. 4.5). The comparison between $MLS_{All}$ and NR reveals almost no bias (<5%) between these two experiments. The standard deviations are lower than 5% at altitudes above (i.e. at pressures lower than) ∼20 hPa and increase toward lower altitude (i.e. when pressure

increases) to values larger than 20% in the tropical upper troposphere. This is the signature of the EnKF noise which is added to each ensemble model state before each analysis step. When considering only MLS daytime observations (the $MLS_{Day}$ run), the assimilation cannot eliminate the biases against NR in the tropical upper troposphere. The relatively poor performances of $MLS_{Day}$ in the South Pole region are due to the lack of observations during the polar night. Both $MLS_{All}$ and $MLS_{Day}$ are losing some correlations against NR in the middle stratosphere. While the weak variability in ozone makes this statistical

indicator less relevant there, the EnKF noise could explain this disagreement.

Based on the comparison between NR, $MLS_{All}$ and $MLS_{Day}$, it turns out that some of the disagreement between AR and NR could be explained. The low number of ALTIUS nighttime observations could partly explain the remaining biases between AR and NR in the tropical upper troposphere but not completely since the biases between AR and NR are still larger than between $MLS_{Day}$ and NR. The EnKF noise could also partly explain the larger standard deviations of the differences between

AR and NR but not completely since $NSD(MLS_{All}$-NR) displays lower values than NSD(AR-NR). Also, at altitudes above (i.e. at pressures lower than) 20 hPa in the Tropics, the standard deviations and the correlations between AR and NR display poorer results than between CR and NR, which cannot come only from the EnKF noise or the low number of ALTIUS nighttime observations.

The remaining discrepancy between AR and NR is due to the larger uncertainty of ALTIUS profiles compared to MLS where

one should take into account the observational error scaling by the Desrozier method. Error scaling profiles calculated by the Desroziers method are given in Fig. 11 for MLS data used in NR and ALTIUS data for the three observation modes used in AR. The error scaling profiles for MLS from $MLS_{All}$ and $MLS_{Day}$ are very similar to those calculated in NR such that they are

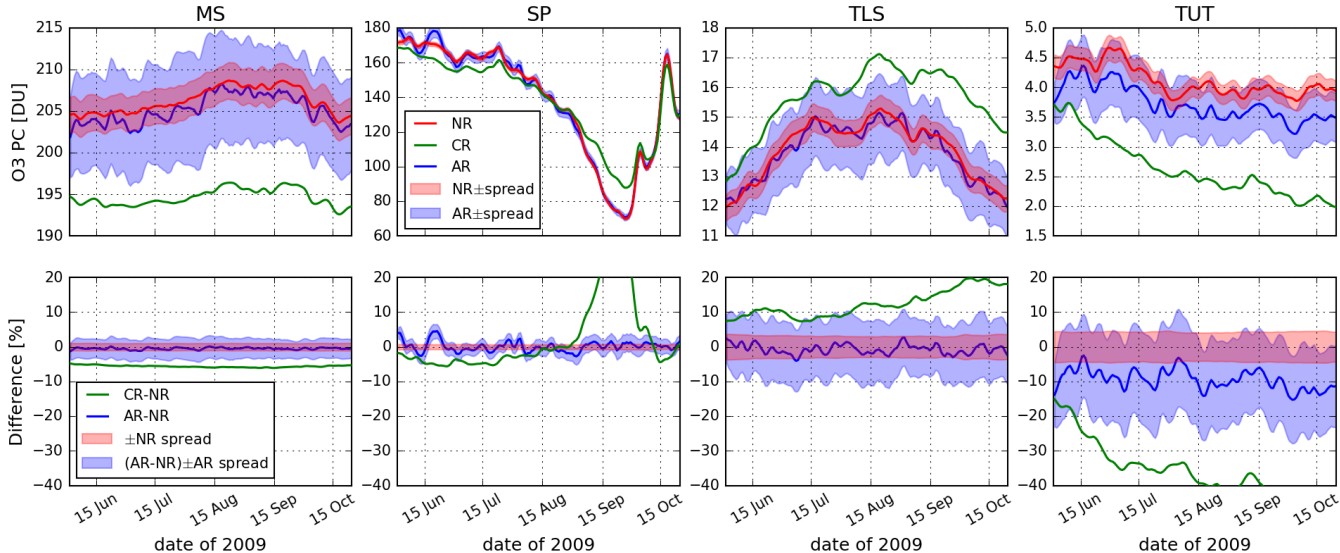

**Figure 12.** Top: Time series of ozone partial column (PC, in Dobson units – DU) of the nature run (NR, red lines), the control run (CR, green lines) and the assimilation run (AR, blue lines) in four regions of the stratosphere: the middle stratosphere (MS, defined within 4-50 hPa and 50°S-50°N), the lower stratospheric South Pole region (SP, within 10-100 hPa and poleward 70°S), the tropical lower stratosphere (TLS, within 70-100 hPa and 25°S-25°N) and the tropical upper troposphere (TUT, within 100-200 hPa and 25°S-25°N). The envelopes around the lines of the nature and assimilation runs correspond to the standard deviations of their respective ensemble state. Bottom: Time series of the differences between {CR,AR} and NR normalized by NR (in %). The red and blue envelopes correspond to, respectively, the NR spread (in %, normalized by NR) and the AR spread around the (AR-NR) line.

not shown. At pressures lower than 20 hPa, where NSD(AR-NR) is larger than NSD(CR-NR), ALTIUS and MLS uncertainties are comparable (see Fig.7c-e) while the error scaling factors for ALTIUS bright limb are two times greater than for MLS. This
is likely due to the propagation of the EnKF noise from NR in the ALTIUS simulated data (see also discussion in Sect. 4.3.2). In the upper tropical troposphere, the scaling factors for MLS and ALTIUS bright limb are comparable and there, it is the larger uncertainty of ALTIUS compared to MLS (see Fig.7c and e) which is responsible for the larger NSD(AR-NR) against NSD(MLS$_{Day}$-NR).

Figure 12 (top row) shows the time series of ozone partial columns from the nature, the control and the assimilation runs in
four regions of the stratosphere: the middle stratosphere (MS, defined within 4-50 hPa and 50°S-50°N), the lower stratospheric South Pole region (SP, within 10-100 hPa and poleward of 70°S), the tropical lower stratosphere (TLS, within 70-100 hPa and 25°S-25°N) and the tropical upper troposphere (TUT, within 100-200 hPa and 25°S-25°N). The second row of Fig. 12 displays the differences between NR and {CR,AR}.

In the four regions, Fig. 12 confirms the time stability of the statistics shown in Fig. 10. In the MS, SP, and TLS, AR captures
the ozone seasonal changes present in NR, something less well achieved by CR. In TUT, AR and NR ozone seasonal changes differ but their agreement is still much better than for the comparison between NR and CR. Being produced by an ensemble Kalman system, the standard deviation of the ensemble state (or spread) of the analyses of NR and AR allows us to measure

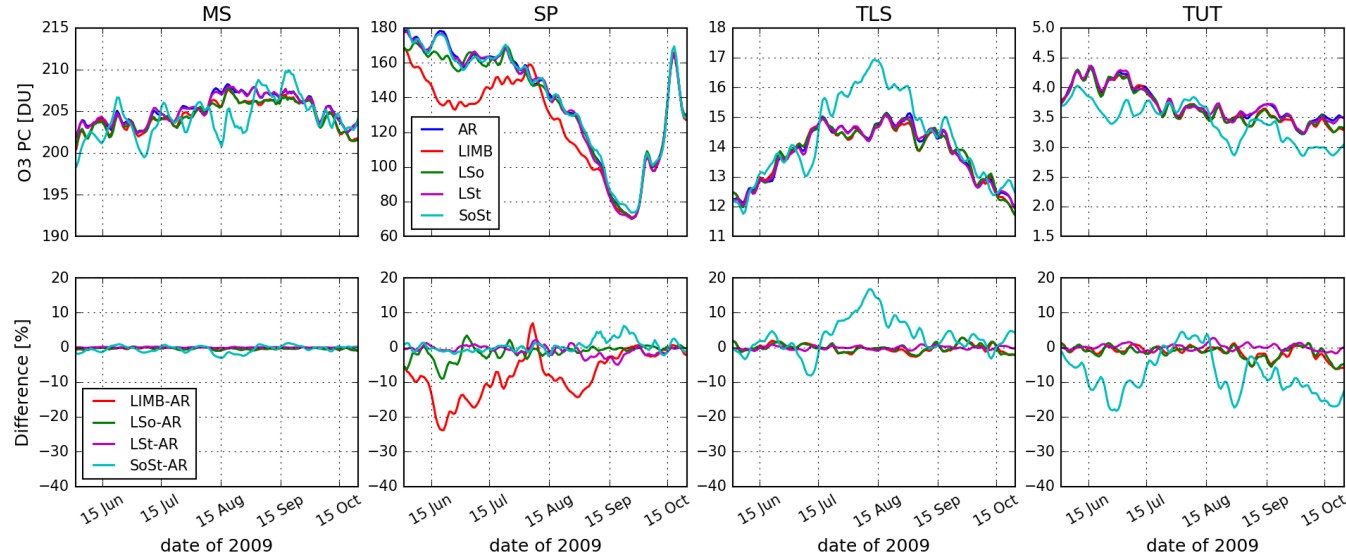

**Figure 13.** As Fig. 12, but showing the assimilation run (AR, blue line), the LIMB run (red line), the LSo run (green line), the LSt run (magenta line) and the SoSt run (cyan line).

the constraint of the assimilated observations (MLS and ALTIUS, respectively) in the analyses. These spreads are also shown in Fig. 12. As expected, the NR spread is smaller than that of AR in all regions.

These results suggest that future ALTIUS profiles will provide ozone analyses unbiased relative to those obtained with MLS from the middle stratosphere to the upper troposphere, polar night included, except in the tropical troposphere. In this region, a reduction in the bias could be obtained using a model with a better representation of tropospheric ozone than in COPCAT. Also, considering the assimilation of nadir ozone total columns in addition to profiles could help to improve the ozone analysis in the upper troposphere. These two improvements will possibly be met in the CAMS system.

## 365    6    Added value of the different ALTIUS modes of observation

This section evaluates the added value of the different ALTIUS modes of observations with four experiments described in Sect. 4.5 (see also Table 2). Figure 13 shows the time series of ozone partial columns from these runs as well as AR in the middle stratosphere, the lower stratospheric South Pole region, the tropical lower stratosphere and the tropical upper troposphere. The differences of these four additional runs with AR are also shown. The SoSt run is relatively far away from the assimilation run

in the TLS and TUT regions but provides analyses relatively close to AR in the middle stratosphere, despite the relatively low number of stellar occultations as compared to bright limb profiles (solar occultations are only measured at high latitudes so they do not constrain the Tropics). Runs using only bright limb observations agree relatively well with AR in the MS, TLS and TUT.

At the South Pole during the polar night, runs using stellar occultations provide the strongest constraint on the system to reproduce AR (LSt and SoSt). Although to a lesser extent, runs using solar occultations also provide a significant constraint (LSo). During the development of the Antarctic ozone hole, starting in September, runs using bright limb data are essential to capture the ozone depletion. Altogether, all ALTIUS modes of observations provide a significant constraint to capture the evolution of ozone during the Antarctic winter.

## 7   Conclusions

The aim of this paper is to evaluate the level of influence of ozone profiles from the ALTIUS UV-VIS-NIR limb sounder to constrain ozone analyses obtained by a data assimilation system. To achieve this goal, an Observing System Simulation Experiment (OSSE) was built using the Belgian Assimilation System for Chemical ObsErvations (BASCOE) and an instrument simulator. Within this OSSE, the nature run used to simulate ALTIUS observations was constrained by ozone profiles observed by the MLS instrument. During the nature run, the model state was saved in the simulated ALTIUS observation space. Profiles in this space were then perturbed using the estimated ALTIUS ozone error covariances to obtain the simulated ALTIUS data. The control run (without assimilation of any data) and the assimilation run (using simulated ALTIUS data) were run at lower resolution than the nature run, but all three runs are using the same advection scheme and ozone chemistry.

Comparisons of the nature run against the control and assimilation runs show that ALTIUS observations will provide significant constraints for the assimilation system in the middle and lower stratosphere, the South Pole region during the polar night and the development of the ozone hole, and in the upper troposphere except in the Tropics. In the tropical upper stratosphere, ALTIUS provides weaker constraints due to the larger uncertainty of ozone profiles in this region and the limited sampling during the night where only a few stellar occultations are available.

Being in a sun-synchronous orbit, ALTIUS will operate in different modes of observation: bright limb during day time, solar occultations at the terminator and stellar/planetary/lunar occultations during night time. Several additional assimilation experiments have been done to evaluate the value of these modes. As expected, bright limb data provide the strongest constraints outside the polar night. During the Antarctic winter and spring, all modes of observation are necessary to constrain the evolution of ozone: stellar occultations are the most important ones during the polar night, followed by solar occultations, while bright limb data are necessary to capture the amount of ozone depletion during the ozone hole period. Despite their relatively low number, stellar occultations also provide a significant constraint in the middle stratosphere, though to a lesser extent than bright limb data.

Overall, this OSSE shows that UV-VIS-NIR instruments like ALTIUS can provide enough information to constrain ozone in chemical data assimilation systems like the Copernicus Atmospheric Monitoring Service (CAMS) system for near real-time ozone analyses and forecasts, as well as for continuing to monitor the stratospheric ozone layer. ALTIUS launch is expected in 2024.

*Author contributions.* QE led the development of the BASCOE data assimilation system, designed its configuration for the ALTIUS OSSE, realized all the BASCOE simulations, produced the figures and wrote most of the text except for the introduction and Sect. 2. EDK simulated the ALTIUS error covariance matrices and led the writing of Sect. 2 and 4.3.2. PhD simulated the geolocation of ALTIUS observations and led the writing of Sect. 4.3.1. All authors participate in the ALTIUS project and have reviewed the paper before its submission.

*Competing interests.* We declare to have no competing interests.

*Acknowledgement.* This research has made use of the SIMBAD database, operated at CDS, Strasbourg, France. The ozone sonde data used in this publication were obtained from the Network for the Detection of Atmospheric Composition Change (NDACC) and are publicly available (see http://www.ndacc.org).

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
