# Peer review of "On the capability of the future ALTIUS UV-VIS-NIR limb sounder to constrain modelled stratospheric ozone"

_Atmospheric Measurement Techniques, 2020_

## Referee Comment (RC1)

**Review of the manuscript "On the capability of UV-VIS limb sounders to constrain modelled stratospheric ozone and its application to the ALTIUS mission" by Errera et al. 2021**

This paper describes and discusses an OSSE experiment designed to assess the impacts of ozone assimilation from ESA's ALTIUS instrument scheduled for launch in 2024. The experiment consists of an MLS ozone analysis performed with a version of the BASCOE system as its nature run, a no-assimilation control and several simulated ALTIUS assimilation runs, where different configurations of bright limb, solar, and stellar occultations are used. The results show that ALTIUS (or a similar) sensor can provide sufficient data to constrain ozone in systems such as CAMS and other reanalyses. This is a very encouraging result.

Performing OSSEs prior to new satellite missions is a standard (in some cases even required) practice that allows one to assess the usefulness of these missions and helps scientists and decision makers plan ahead. That makes the topic an important one and in line with the scope of AMT. It is great to see that our community will have an alternative ozone data source in addition to the OMPS series in the dreaded but inevitably approaching post-MLS era. The paper is a pleasure to read. The science is sound, the manuscript is clearly and logically written; the figures are legible, and the supplementary information and plots are helpful. I have only a handful of specific and technical comments and suggestions for minor revisions. Once these are addressed the paper should be ready for publication.

**Specific comments**

L41. "limb-scattered solar light during daytime" is what's called "bright limb observations" below, right? If that's the case, I'd suggest introducing this term here and using a consistent terminology throughout the paper.

LL79-82. There have been more solar occultation sensors than the ones mentioned here: HALOE, POAM, ILAS. I feel it should be mentioned here that solar occultation is a well-established measurement technique for ozone.

Table 1. Please explain what threshold uncertainty is. I take it to mean the worst but still acceptable uncertainty. Is that correct?

L101. There's a new paper now in ACPD that discusses this scheme (at least I think it's the same one): https://acp.copernicus.org/preprints/acp-2020-1261/#discussion; the authors may want to consider citing it if the new Monge-Sanz paper is accepted by ACP before this one is.

L108. How well are the effects of polar heterogeneous chemistry on ozone represented?

L126. I take it that observation uncertainties are assumed to be uncorrelated (which is fine). Is that correct?

L141. "15 days later". Is it because that way the initial condition has sufficiently departed from the assimilated state in NR?

L151. The latest version of the V4.2 data quality document is from 2020 (Livesey et al., 2020). I don't believe the recommendations of estimated uncertainties for ozone changed between 2015 and 2020. If they didn't it would be better to cite the latest available document (https://mls.jpl.nasa.gov/data/v4-2\_data\_quality\_document.pdf). Same applies for all the other instances this document is cited.

L168. In the upper stratosphere the chemical time scales for ozone drop rapidly with altitude making data assimilation a particularly hard problem. Can you explain the decision to assimilate ozone at those pressures, vis a vis Errera et al. 2019 who imposed a cutoff pressure at 4 hPa for ozone despite using a much more advanced chemistry model? I'm not saying I disagree with that decision, I'm just curious if the authors find the results in the upper stratosphere meaningful.

L174. Uncorrelated with each other or with the NR?

L185. It would make sense to re-emphasize that CR is also driven by ERA-Interim so that the meteorology is consistent with NR.

LL217-252. I like it that the authors provided this extensive explanation. It really helps if the reader knows what was done and why. It looks like you found the right balance between what needs to be done in the way of simulating ALTIUS and what can be done. I appreciate it.

LL240-243 and Figure 7. The MLS data quality document also contains accuracy estimates (reported as 2-sigma). The plot only shows precisions. Is it because the estimated ALTIUS error standard deviations also represent precision? Is there anything we can say (perhaps the answer is "no") about the expected accuracy of ALTIUS ozone data?

L272 and Fig. 9. This harkens back to my question about the ability of the chemistry scheme to represent polar ozone loss. It looks like CR misses it completely. Can you explain? I understand that this is somewhat tangential to the topic of this paper, so just a couple of sentences of explanation will be enough.

~L292. While the difference standard deviation is reduced nicely for the most part in AR compared to CR, there is a patch of values <5% in CR-NR between 30°S and 60°N at p $<\sim20$  hPa that becomes slightly worse (>5%) in AR-NR. Can the authors comment on that?

L294-296. Why not add hatching to the plot to show the regions of significance? Or is the improvement significant everywhere?

LL302-304. Even though AR sits within or not too fat from the NR envelope, it shows a lot more variability than NR on daily-to-weekly scales, especially in MS md TLS. I'm looking at the jagged blue line compared to the red line. Can the authors explain what's happening there?

LL307-308. Would it be possible to test this? One could run an additional short experiment with double the number of simulated "ALTIUS" observations, e.g. by also simulating observations along the descending night node. Those wouldn't, of course, literally make sense as simulated ALTIUS data but looking at what this does to the spread could help substantiate this claim.

L355. I would be more specific here: "to constrain **ozone** in chemical **data assimilation** systems such as"

**Suggestions for technical corrections**

L10 and below. I would replace the word "weight" with "impact" ("impact on the analysis"). I reserve "weight" to situations where we talk about different weights given to different data sources in data assimilations, etc. It's just my preference. Up to the authors.

L18 "signature". I think something like "signing" or "implementation" would sound better.

L24 "limb-looking"; I suggest "limb-viewing" instead.

L25. I suggest avoiding using ellipses ("…") in a scientific text. This sentence (causes affecting the ozone layer and their consequences) doesn't read well. Please, revise.

L38. Nearly polar  $\rightarrow$  near polar

L40 part  $\rightarrow$  parts

LL67-71. Please use either "Section" or "Sect." consistently; I would opt for the full word "Section".

L100 "throughout"  $\rightarrow$  "through" or "via"

L112 "consists in"  $\rightarrow$  "consists of"

L93. In the interest of being concise, I don't think it's necessary to repeat what the three geometries are, given they were just discussed in detail. I suggest deleting the text in the parentheses.

L286. "reduced below"  $\rightarrow$  "reduced **to** below"

L334. Again, I don't think "weight" is the best word here. Perhaps "capability" would work better, or "impact on ozone analyses" or something like that.

Kris Wargan

---

## Author Comment (AC1)

**Responses to referee#1 (Kris Wargan)**

We would like to thank referee#1 for his constructive review. Our responses are organized as follows. We first summarized the major modifications in the manuscript according to the comments of the referees then we provide detailed responses to each referee's point. Our responses are written in italic green.

**Major modifications in the manuscript**

- 1. The title has been revised in order to be less general, according to comments from referees 2 and 3 and is now: "On the capability of the future ALTIUS UV-VIS-NIR limb sounder to constrain modelled stratospheric ozone"
- 2. We also realized that the ALTIUS simulated profiles have inherited some noise from NR since the EnKF procedure adds small perturbations to each model state at each model time step. This would explain some of the larger variability and lower correlation in the comparison between (AR,NR) and (CR,NR) which was pointed out by referee 1 & 2. This is now discussed in the paper with the help of the error scaling profiles calculated by the Desroziers method for NR and AR which are shown. This is something which has not been anticipated before the first submission of the manuscript.
- 3. Two additional experiments have been added in order to evaluate the impact of the low sampling of ALTIUS during the night, a questions raised by referee 1. They consider the assimilation of MLS all data and MLS daytime data with a system configured as for the assimilation run.

**Responses**

This paper describes and discusses an OSSE experiment designed to assess the impacts of ozone assimilation from ESA's ALTIUS instrument scheduled for launch in 2024. The experiment consists of an MLS ozone analysis performed with a version of the BASCOE system as its nature run, a no-assimilation control and several simulated ALTIUS assimilation runs, where different configurations of bright limb, solar, and stellar occultations are used. The results show that ALTIUS (or a similar) sensor can provide sufficient data to constrain ozone in systems such as CAMS and other reanalyses. This is a very encouraging result.

Performing OSSEs prior to new satellite missions is a standard (in some cases even required) practice that allows one to assess the usefulness of these missions and helps scientists and decision makers plan ahead. That makes the topic an important one and in line with the scope of AMT. It is great to see that our community will have an alternative ozone data source in addition to the OMPS series in the dreaded but inevitably approaching post-MLS era. The paper is a pleasure to read. The science is sound, the manuscript is clearly and logically written; the figures are legible, and the supplementary information and plots are helpful. I have only a handful of specific and technical comments and suggestions for minor revisions. Once these are addressed the paper should be ready for publication.

**Specific comments**

L41: "limb-scattered solar light during daytime" is what's called "bright limb observations" below, right? If that's the case, I'd suggest introducing this term here and using a consistent terminology throughout the paper. *Done.*

L79-82: There have been more solar occultation sensors than the ones mentioned here: HALOE, POAM, ILAS. I feel it should be mentioned here that solar occultation is a well-established measurement technique for ozone.

These instruments are omitted because they are not working in UV-Vis-NIR wavelengths. This will be clarified.

Table 1: Please explain what threshold uncertainty is. I take it to mean the worst but still acceptable uncertainty. Is that correct? *This is indeed the meaning, added in the table caption.*

L101: There's a new paper now in ACPD that discusses this scheme (at least I think it's the same one): https://acp.copernicus.org/preprints/acp-2020-1261/#discussion ; the authors may want to consider citing it if the new Monge-Sanz paper is accepted by ACP before this one is.

This paper is still under review so it is not cited.

L108: How well are the effects of polar heterogeneous chemistry on ozone represented? In L104-105, it is stated that "..., COPCAT chemistry has the advantage of providing a better representation of polar ozone depletion... (Monge-Sanz et al., 2011, Jeong et al. 2016)." *Please, refer to these two papers for a exhaustive evaluation (note that we added Monge-Sanz et al. here).*

L126: I take it that observation uncertainties are assumed to be uncorrelated (which is fine). Is that correct? Yes, this is discussed in Desroziers et al.

L141: "15 days later". Is it because that way the initial condition has sufficiently departed from the assimilated state in NR?

Yes. The sentence has been rewritten to clarify this point: "It is running with the same spatial resolution than CR and is initialized 15 days later than NR and CR with the  $O_3$  state from CR as initial conditions (the 15-day delay allows AR to have initial conditions sufficiently departed from NR initial conditions)."

L151: The latest version of the V4.2 data quality document is from 2020 (Livesey et al., 2020). I don't believe the recommendations of estimated uncertainties for ozone changed between 2015 and 2020. If they didn't it would be better to cite the latest available document (https://mls.jpl.nasa.gov/data/v4-2\_data\_quality\_document.pdf). Same applies for all the other instances this document is cited.

Livesey et al. (2015) has been changed by Livesey et al. (2020).

L168: In the upper stratosphere the chemical time scales for ozone drop rapidly with altitude making data assimilation a particularly hard problem. Can you explain the decision to assimilate ozone at those pressures, vis a vis Errera et al. 2019 who imposed a cutoff pressure at 4 hPa for ozone despite using a much more advanced chemistry model? I'm not saying I disagree with that decision, I'm just curious if the authors find the results in the upper stratosphere meaningful.

It is true that ozone assimilation with BASCOE full chemistry can lead to several issues above ~4 hPa. First, the system cannot completely eliminate the FmO bias for ozone. Moreover, in 4D-Var, the system used to introduce negative biases in several species (e.g. HCl, NOx, H2O) to increase the amount of ozone. This the reason why several years ago, we have decided to not assimilate ozone in the upper stratosphere. However, several studies using ozone linearized chemistry displayed good results for ozone assimilation up to 1 hPa (e.g. Geer et al., 2006, ACP). According to the FmO statistics shown in our paper (e.g. Fig. S1 and S2), this seems not to be the case for the BASCOE COPCAT scheme. The reason is likely due to a negative bias in the O3 climatology of COPCAT, this climatology being computed based on multiannual simulation of the SLIMACT CTM, which suggests a negative bias in SLIMCAT as well. This has not been included in the paper where we think it is out of its scope.

**L174: Uncorrelated with each other or with the NR?**

*Obviously, uncorrelated with each other. This is clarified as: "… ozonesondes (Figs. 3 and 4). The uncertainties of ozonesondes are assumed…"*

L185: It would make sense to re-emphasize that CR is also driven by ERA-Interim so that the meteorology is consistent with NR.

Done as follows. "The control run (CR) is based on a BASCOE free model simulation (no assimilation) also driven by ERA-Interim but with a lower horizontal resolution than in NR:..."

L217-252: I like it that the authors provided this extensive explanation. It really helps if the reader knows what was done and why. It looks like you found the right balance between what needs to be done in the way of simulating ALTIUS and what can be done. I appreciate it.

**Thank you for this positive comment.**

L240-243 and Figure 7: The MLS data quality document also contains accuracy estimates (reported as 2-sigma). The plot only shows precisions. Is it because the estimated ALTIUS error standard deviations also represent precision? Is there anything we can say (perhaps the answer is "no") about the expected accuracy of ALTIUS ozone data? Satellite instruments are calibrated before the flight to expect no bias. So remaining biases are unexpected and are difficult to evaluate before real observations are validated. This is

why the MLS and ALTIUS accuracy is not discussed here.

L272 and Fig. 9: This harkens back to my question about the ability of the chemistry scheme to represent polar ozone loss. It looks like CR misses it completely. Can you explain? I understand that this is somewhat tangential to the topic of this paper, so just a couple of sentences of explanation will be enough.

In fact, the CR run is not so bad in representing the polar ozone loss. From Fig. 4, we could see that the agreement with ozonesondes is relatively good (see also Fig. 11 for the SP). The profile in Fig. 9 is shown on Sept. 15 when the ozone loss is the most important and the timing of the loss in CR seems to be delayed from NR and AR. This is clarified by adding at the end of Sect. 4.2. : "However, note the relatively good representation of Antarctic ozone depletion in CR, thanks to the COPCAT chemistry, when compared to ozonesondes (see Fig. 4)."

~L292: While the difference standard deviation is reduced nicely for the most part in AR compared to CR, there is a patch of values <5% in CR-NR between 30°S and 60°N at p<~20 hPa that becomes slightly worse (>5%) in AR-NR. Can the authors comment on that? This issue has been studied in more details in the revised manuscript. The reason is due to the noise that ALTIUS simulated data have inherited from NR. This is something we did not anticipate we the manuscript was submitted.

L294-296: Why not add hatching to the plot to show the regions of significance? Or is the improvement significant everywhere?

The improvement is significant almost everywhere and adding a hatching would degrade the figure. So the figure has not been updated.

L302-304: Even though AR sits within or not too far from the NR envelope, it shows a lot more variability than NR on daily-to-weekly scales, especially in MS md TLS. I'm looking at the jagged blue line compared to the red line. Can the authors explain what's happening there?

Again, this issue has been studied in more details in the revised manuscript and would be attributed to the EnKF noise that has been inherited by the ALTIUS simulated profiles.

L307-308: Would it be possible to test this? One could run an additional short experiment with double the number of simulated "ALTIUS" observations, e.g. by also simulating observations along the descending night node. Those wouldn't, of course, literally make sense as simulated ALTIUS data but looking at what this does to the spread could help substantiate this claim.

This sentence has been removed according to our response to the previous point. However, it was much easier to assimilate MLS daytime data only instead of trying to simulate ALTIUS nighttime limb profiles from which SNR table are not available.

L355: I would be more specific here: "to constrain ozone in chemical data assimilation systems such as" *Done*

Suggestions for technical corrections

L10 and below: I would replace the word "weight" with "impact" ("impact on the analysis"). I reserve "weight" to situations where we talk about different weights given to different data sources in data assimilations, etc. It's just my preference. Up to the authors. *"Weight" has been replaced by "impact".*

L18: "signature". I think something like "signing" or "implementation" would sound better. "Signature" replaced by "implementation".

L24: "limb-looking"; I suggest "limb-viewing" instead. *OK, done.*

L25: I suggest avoiding using ellipses ("...") in a scientific text. This sentence (causes affecting the ozone layer and their consequences) doesn't read well. Please, revise. I did not find something better, so the sentence has not been changed.

L38: Nearly polar -> near polar *Done.*

L40: part -> parts Done.

L67-71: Please use either "Section" or "Sect." consistently; I would opt for the full word "Section".

This is the AMT standard. I quote here from https://www.atmospheric-measurement- techniques.net/submission.html#manuscriptcomposition "The abbreviation "Sect." should be used when it appears in running text and should be followed by a number unless it comes at the beginning of a sentence."

L93: In the interest of being concise, I don't think it's necessary to repeat what the three geometries are, given they were just discussed in detail. I suggest deleting the text in the parentheses.

Done.

L100: "throughout" -> "through" or "via" "Throughout" replaced by "via".

L112: "consists in" -> "consists of" *Done.*

L286: "reduced below" -> "reduced to below" *Done.*

L334: Again, I don't think "weight" is the best word here. Perhaps "capability" would work better, or "impact on ozone analyses" or something like that. "Weight" has been replaced by "impact".

---

## Author Comment (AC2)

**Responses to referee#2 (Yves Rochon)**

We would like to thank referee#2 for his constructive review. Our responses are organized as follows. We first summarized the major modifications in the manuscript according to the comments of the referees then we provide detailed responses to each referee's point. Our responses are written in italic green.

**Major modifications in the manuscript**

- 1. The title has been revised in order to be less general, according to comments from referees 2 and 3 and is now: "On the capability of the future ALTIUS UV-VIS-NIR limb sounder to constrain modelled stratospheric ozone"
- 2. We also realized that the ALTIUS simulated profiles have inherited some noise from NR since the EnKF procedure adds small perturbations to each model state at each model time step. This would explain some of the larger variability and lower correlation in the comparison between (AR,NR) and (CR,NR) which was pointed out by referee 1 & 2. This is now discussed in the paper with the help of the error scaling profiles calculated by the Desroziers method for NR and AR which are shown. This is something which has not been anticipated before the first submission of the manuscript.
- 3. Two additional experiments have been added in order to evaluate the impact of the low sampling of ALTIUS during the night, a questions raised by referee 1. They consider the assimilation of MLS all data and MLS daytime data with a system configured as for the assimilation run.

**Responses**

The study focuses on estimating the potential impact of ALTIUS ozone observations in improving model short-term forecasts and assimilation analysis of the upper troposphere to middle atmosphere ozone field over the globe using an OSSE. The applied approach and evaluation are scientifically sound. Generally, the content is clearly presented, concise, and well organized. On occasion, it would benefit from additional information as indicated below.

Some reframing is recommended, starting with the title. See also related specific comments in the introduction and elsewhere below.

There are minor issues in the grammar in some sections, some of which were considered not worth pointing out. There is the issue of choosing the singular vs plural with 'bias', standard deviation', and 'correlation' especially. The plural would be better in many cases and the choice between the two could be more subjective in others. It would need to be plural when referring to quantities (potentially multiple values).

While the suggested revisions are not very major, the paper would benefit from another review.

Specific comments

Title: "UV-VIS-NIR" instead? *Done.*

Title: 'On the capability of UV-VIS limb sounders to ...' is too general in part considering that the content and text of the paper referring essentially to the ALTIUS mission and instrument specifications – even though some implications would admittedly hold anyways to other limb sounders. Many previous studies have already provided some insight on different aspects of this topic. I recommend having the title not refer to 'the capability of UV-VIS limb sounders' and focus on the capability of ALTIUS itself.

A new title is suggested, see the introduction of this document.

L19: Providing a reference would be good. *WMO (2018) has been added.*

L31: Please provide a related reference for the first sentence if possible. *Fussen et al. (2019) has been moved from two sentences later here.*

L38-41: Any reference or related document? *Fussen et al. (2019) is now cited.*

L48: Include also the mention and references of OSSES regarding stratospheric chemical composition (including ozone). This is not the first OSSE involving stratospheric ozone measurements.

To the best of our knowledge, this paper is the first discussing OSSE for a satellite instrument dedicated to stratospheric ozone profile measurements. We have found some conference abstracts, powerpoint presentations or ESA PREMIER report title (but note the report itself) which are all difficult to cite in a peer reviewed. We have added at the end of the §: "To the best of our knowledge, this paper is the first discussing an OSSE for a satellite instrument dedicated to stratospheric ozone profile measurements."

As well conclusions from previous assimilation studies involving actual satellite ozone data (including profile sources such as MLS and others) would also be relevant to this work. It would be important to summarize/mention relevant conclusions from earlier assimilation studies involving simulated and actual ozone measurements from satellites in relation to the objectives of this study.

Done where we also mention that MLS measures during day and night.

What conclusions from earlier OSSEs and OSEs with satellite ozone data are pertinent to this study and what might all of these lack in answering questions regarding the impact of ALTIUS (this relates to one or two statements in Section 2)?

Again, we don't know any previous OSSEs or OSEs studies focusing on stratospheric ozone measurement from space.

L55-56: Specifying BASCOE without COPCAT can be misleading as most already familiar with BASCOE may/will assume use of the full chemistry package at this point. Maybe best to mention either both or neither here.

The reference to BASCOE has been removed.

L56: Starting with the mention of BASCOE in the sentence lends to confusion in reading the remainder of the sentence. It is recommend to instead have a paragraph or sentence before L55 to describe/mention the need for a simulation process to provide the measurements used for investigating both questions. *This has been done.*

L55-65: Then again, some/much of the content here would fit better in an introduction of a methodology section (e.g. Section 4) instead of the introduction. Some of the content summarizes the methodology as opposed to introducing the subject. Removing some of the methodology details in these paragraphs (if not most these entirety of these two paragraphs) is recommended.

We do not completely agree with the referee, where we believe that methodology can be presented in the introduction. However, the § has been rewritten in order to improve its readability.

L116-117: 'As well, the effective vertical resolution stemming from the averaging effect of the averaging kernels - ... - so that their use' Is this what is meant? (If so, does this actually apply to the ALTIUS ozone product – as this is dependent on the applied retrieval constraints, including the relative effects of the measurement and a priori/constraint error covariances/weights.) Otherwise, the last part of the sentence does not seem to work. *We apologize but we do not understand the comment.*

L120: Please state the assimilation window period and/or interval (e.g. every 6 hours covering +/- 3 hours about each synoptic time?)

Data are assimilated at every model time step, as stated in Skachko et al. (2014). We added in the § starting with: "Two data assimilation methods are available in BASCOE, ... When available, observations are assimilated at every model time step..."

L128: 'using a minimum of three' "Using at least three" has been used.

L131: In OSSEs, one could potentially or often simulate many or all observation sources – this depends on the intent and the setup. So the control run could technically, depending on the setup and what is intended, use simulated observations except for one or more target sets. Maybe some re-phrasing is needed.

The sentence has been rewritten as: "The control run, being either a free model run or a data assimilation run without (all or a part of) the simulated observations, ..."

L134-135: 'This ensures .. only ...' – maybe not likely. All the 'old instruments' and the common aspect of model physics, etc, would contribute some similarity in results. Please re-phrase. Maybe the intent was to say that the 'increased similarity or agreement between the NR and AR results "as compared to that between the NR and CR" is most likely due to ...' "Only" has been replaced by "most likely".

L137: Saying it is 'solved' may be too strong. Maybe something like 'the concern of the identical twin issue is largely removed' or attenuated (maybe not entirely removed). We understand the concern of the reviewer but in the context of the sentence, we still believe that "solved" is the best choice. Other possibilities would be "settled" or "fixed". "Removed" or "attenuated" would not work here e.g. "All runs use the same BASCOE model (see Sect. 3) and the identical twin problem is attenuated as follows."???

L139: If the CR or another AR does not perform assimilation of other ozone profile sources such MLS or OMPS-LP (NPP) for example, then the target AR will not show the value added benefit of including ALTIUS to one or more other ozone profile sources. Some mention of not doing so would be relevant here - if not also in the conclusions sections. We added at the end of the §: "While OSSE will in general measure the value of a new instrument added in an existing observing system, our goal is more to measure how the new

instrument (i.e. ALTIUS) could replace an old one (i.e. MLS). This is why the control and the assimilation runs do not assimilate MLS data."

Figures 1, 3, and 4: Having AR and CR results in the NR section is not ideal. Maybe the text in section 4.1 should indicate that the AR and CR results included in the figures will be discussed in later sections.

Done as follows: "The success of MLS assimilation is verified by means of the Chi2-test and the forecast minus observation (FmO) statistics (results of CR and NR shown in the following figures will be discussed in later sections)."

Additionally, comparing AR and CR to actual measurements, other than MLS maybe as it was assimilated for the NR, is not as meaningful or clean as comparing to the NR itself – since the NR is the truth for the AR and the CR (even MLS itself is not the truth here – it is the NR). *This is now on top of Sect. 5 when comparing AR with independent data (see also our response to the point L305-314).*

It would be worth discussing this. It is good to compare the NR with these observations though to evaluate the realism of the NR and the effectiveness of the MLS assimilation as is done in this section.

Figure 3: Why do correlations reach 1.0 (or nearly 1.0) at upper levels? Is it truly because of consistency or something else? It would be good to provide an explanation somewhere in the text.

We thank the reviewer for its question which helps us to spot an issue in the code used to make the Fig. 3. This issue has been fixed (removing ozonesondes data above 10 hPa because too few soundings are going above that altitude) and the figure has been reprocessed.

Figure 3: Why are mean differences and std. dev. often smaller or near zero above or as about 10hPa? One would/might instead expect larger values at upper levels. *Issue fixed, see our response to the previous point.*

L162: State for which forecast F time(s)/periods (e.g. +/- 3 hours about each synoptic/analysis time/period)?

As mention in Sect. 3, available observations are assimilated at each model time step. So forecast period is 30'. This is discussed in Skachko et al. (2014) and the reader interested in technical details on the BASCOE-EnKF should refer to this paper.

L165: If these are actually at the times of each assimilation, the statistics may show even more so the assimilation effectiveness of BASCOE, i.e. a sanity check on the assimilation of MLS, when comparing to MLS.

We are sorry, we are not sure to understand the reviewer's comment.

L166: How are the correlations calculated? Are these anomaly correlation coefficients (if so provide a reference)? If not, are the denominator standard deviations those prescribed for the forecasts via the ensembles (and observations when involving observations). Please provide some specifics.

We have calculated the sample correlation coefficients as:  $CORR(x,y) = [SUM_i (x_i-x_{mean})(y_i-y_{mean})]/[SQRT(SUM_i (x_i-x_{mean})^2 SUM_i (y_i-y_{mean})2)].$ "Sample correlation coefficients" now replaced "correlations" in the relevant figures of the revised manuscript.

L166: Specify vertical ranges of applicability and temper the comment with 'typically', 'mostly' or .,. when these limits are not satisfied for all latitude bands. The sentence 'These are, however, ...' would not be needed in that case – and it is better for the previous sentence to be precise in its statements.

Vertical range of the TTL added.

Section 4.2. The control run

See earlier comments related to Figures 3 and 4 and the NR being the truth for the CR and AR.

Section 4.3.1 Profile geolocation

L210: It might be relevant to mention somewhere that MLS provides limb profiles for both day and night conditions in the event some readers may not be familiar with MLS, hence the larger number of MLS measurements. (maybe this was done earlier in the paper?) *This was already the case, see L149-150 saying "MLS measures vertical profiles of around fifteen chemical species, including ozone, during day and night." This has also been added in the introduction.*

**Section 4.3.2. Ozone profiles**

I did not notice any mention of the spatial sampling (for the daytime limb measurements) and especially the vertical resolution(s) of the ALTIUS profiles. If not mentioned, it is important to do so.

ALTIUS vertical resolution (1-2 km for solar occultation and 2-3 km for other observations mode) in now mentioned in Sect. 2. The sampling along track for bright limb data is around 200 km, this information has been added in Sect. 4.3.1.

L218-220: There was no mention of accounting for the geometry of the measurements (limb viewing) and consideration of scattering not just within a vertical column.

This paragraph discusses how a full-blown implementation should look like. With this approach, radiative transfer calculations should be done for every simulated observation. It is clear that a limb-scattering code (with multiple scattering capabilities) is needed to simulate the bright limb observations, whereas a simpler transmittance code is needed for the occultations.

L228: This is done to generate the covarirance matrix. It needs to be stated before this sentence, saying that covariance matrices are defined first before simulating the final observation profiles.

The text has been updated as follows: "To overcome this issue, a sample of L2 error covariance matrices have been calculated for a number ozone profile conditions representative of the OSSE period (June-October 2009). These matrices will then be used to perturb the NR state saved in the ALTIUS space, thus providing the ALTIUS simulated profiles. The error covariance matrices are obtained by linear propagation..."

L231 (or below): 'linear interpolation from the NR to the ALTIUS altitudes'? What are the vertical resolutions? (Do I just miss this in the earlier text?) *The ALTIUS vertical resolution is now given in Sect. 2.*

L246 (and earlier): What is the ALTIUS space? (resolutions) How was interpolation done? (linear?) Was limb geometry considered in the interpolation? Normally, the latter is not done for simplicity.

It is now defined in the Sect. 4.3.1: "... bright-limb measurements resume (when sza<85°). This procedure provides the geolocation of latitude, longitude and time of ALTIUS simulated

profiles at the 30 km altitude tangent point. Variation of the latitude/longitude/time with the altitude of the tangent point has not been taken into account. At each ALTIUS geolocation, a profile from 0 to 100 km with a step of 1 km is considered. The ALTIUS space is thus defined by the one-dimensional vertical grid and the one-dimensional time/latitude/longitude vectors."

Sections 4.4 and 4.5: Combining the two sections into one is recommended, i.e. 'The assimilation runs'

This has not been done because two additional experiments using MLS observations have been included.

L253: 'A reference assimilation run ... assimilation of all ALTIUS' (or 'reference' replaced by another preferred word) *Done.*

L262: 'as the reference assimilation' *This sentence no longer exists.*

Section 5. Evaluation of the 'Reference Assimilation 'Run' The section title has not been changed. In our mind, there only one assimilation run which is AR.

L264: 'and the reference assimilation' *Not done.*

L290 or so: Is the COPCAT more chemically fast acting in the upper troposphere than in the stratosphere (as it would/might also be at even higher vertical levels)? *This is a possibility that we did not verify.*

L305-314: This discussion is rather late in the text considering that this pertains to Figs. 3 and 4. As a reader, I was a bit puzzled not seeing this near the beginning of the section. See also earlier points regarding Figs. 3 and 4 in comparing to actual observations in this case. *This § has been moved in Sect. 4.4 with some text adjustment.*

L310-314: Please discuss cases where CR has smaller standard deviations than for AR in Figure 4.

Done, see our reply to reviewer #1.

L317: Maybe not so evident (if not that likely). Any demonstrable proof from other work (e.g. from CAMS itself)? If so, a reference would be good. If not, maybe better to exclude that statement.

This is a hypothesis, I wrote "could". I think the authors could also elaborate around their results so I would like to keep this sentence. Nevertheless, we changed "These two improvements will likely be met in the CAMS system" by "These two improvements will possibly be met in the CAMS system". (I.e. "likely"=>"possibly").

Section 6. Added value of the different ALTIUS modes of observation

Section 7. Conclusions

L334-335: Suggest removing the commas or re-writing to refer specifically to ALTIUS, e.g. 'from the ALTIUS UV-VIS-NIR limb sounder' which would be better. *Done.*

L335: 'analyses'. So are the F of FmO analyses? Will be changed from "to constrain ozone analyses", by "to constrain modelled ozone"

L348: 'Several assimilation experiments' or 'A few assimilation experiments' For us, "Several additional assimilation experiments" is better so the text has not been updated.

Technical corrections:

L5: 'limb-scattered'. As well the paper also uses 'bright-limb' and 'bright limb'. "Bright limb" is now the term used in the revised manuscript.

L31-32: Proposed and supported by agencies (of countries) and not the countries themselves (minor point though) *Well, indeed, a minor point.*

L34-35: 'with a latency of less than 3 hours from the sensing to the retrieval product delivery to operational services' *Done.*

L38: The removal of comma is suggested as it cuts the actual phrasing. *Done.*

L41: ': measurements ... ' Done.

L45: 'systems using data assimilation' *Done.*

L53: 'measurements) in particular' (remove comma) or 'measurements) with some emphasis on the polar night' *Comma has been removed.*

Section 2. The ALTIUS Mission

Section 3. The BASCOE System

L113: It is worth referring to FGAT (first guess at appropriate time) here?

BASCOE has never referred to FGAT so far. Data around the model time step +/- half the model time step are assimilated, as stated in Skachko et al. (2014). Please, see also our previous replies above.

L113: 'It is used to save' *Done.*

L115-116: 'Averaging kernels have not been applied in this study since the BASCOE ...' *Done.*

L119-120: 'in BASCOE, the four-dimensional variational method (....) and the EnKF ...' *Done.*

L123: "using Desrozier's method" or "using the Desrozier method" The later choice was adopted. (Desroziers is spell with an ending "s".)

L123: 'based on requiring' instead of ', allowing to have'

No, the Desroziers method is not based on requiring a Chi2 close to 1. Having a Chi2 close to 1 is due because the hypothesis used in Desroziers are met. "Allowing to have" has been kept.

L124: 'model forecasts weighted' *Done.*

L125: 'covariances; m\_k is the number ...' *Done.*

L136: 'setup' as used elsewhere. *Done.*

L146: Remove or re-phrase the added ', done, detailed hereafter as well'. Rephrased as: "Several additional experiments are also summarized in Table~\ref{exp} and detailed at the end of this section."

Section 4.1. The nature run

L148: 'the BASCOE ...' *Done.*

L151: 'given in Livesey et al. (2015).' *Done.*

L166: biases, standard deviations, and correlations (plural) *Done.*

Section 4.2. The control run

Section 4.3.1 Profile geolocation L195: 'than a 1.5 visual magnitude' or 'than a visual magnitude of 1.5' *The later choice implemented.*

L206: 'tangent altitudes of the lines of sight' *Done.*

Figure 7: 'The color code indicates' ... 'error standard deviation profiles correspond to' (twice) ... 'MLS single profile precision error standard deviations' (also plural) ... 'with a range of' .... 'is shown in panels ...' *Done.*

Section 4.3.2. Ozone profiles

L218-220: 'would be used', 'would then be used', 'data would undergo', 'would be retrieved' *Done.*

L228: 'a set of sample ozone profiles ... was produced.'? *Done.*

L231: 'sample set' *Done*.

L234: 'for bright' *Done.*

Sections 4.4 and 4.5: Combining the two sections into one is recommended, i.e. 'The assimilation runs'

L261: Table 2 has SuSt and not SoSt. *SuST replaced by SoST in the table.*

Section 5. Evaluation of the 'Reference Assimilation 'Run'

L269: 'agreements and differences between the NR and the CR and AR is provided' (or '... NR and {CR,AR} ...') *Done.*

L270: 'Above the South' *Done.*

L273: 'underestimates' *Done.*

L277-278: 'biases', 'standard deviations' and 'correlations' Done. ("correlations" replaced by "sample correlation coefficients".) L278: 'mean values' *Done.*

L282-...: Better to use plural form again maybe unless referring to the concepts of bias, ... (including Fig. captions) *Done.*

L294: 'such cases' or 'such a case' ... 'Finally, we have also checked ...' "Such a case" used. "Finally, we have also checked" implemented.

Figure 11: 'runs', 'standard deviations', 'red and blue envelops' – if not others. Section 6. *Done.*

Added value of the different ALTIUS modes of observation

L323: 'differences .... are also shown' *Done.*

L325: Remove 'remember that' Also, how about 'limb profiles; solar ....' instead "remember that" has been removed.

L326: 'using only bright limb' *Done.*

L328: 'At the South Pole' *Done.*

L329: 'LSt' *Done.*

Section 7. Conclusions

L334: 'evaluate the level of influence of' *Done.*

L337-339: Use past tense instead of present tense. *Done.*

L340-341: 'assimilation runs' .... ', and all runs used the' As mentioned earlier, in our paper, there is only one "assimilation run" which is AR. Other runs are called "assimilation experiment". So the singular has been kept.

L347 and earlier/elsewhere: alternating use of 'bright-limb' and 'bright limb'. Consistency would be preferred. *Done.*

Acknowledgments

L363-364: As the source of ozonesondes is provided, how about for MLS as well (if not others)?

NDACC request to be acknowledged, not MLS, MIPAS or ACE-FTS data providers.

---

## Author Comment (AC3)

**Responses to referee#3**

We would like to thank referee#3 for his constructive review. Our responses are organized as follows. We first summarized the major modifications in the manuscript according to the comments of the referees then we provide detailed responses to each referee's point. Our responses are written in italic green.

**Major modifications in the manuscript**

- 1. The title has been revised in order to be less general, according to comments from referees 2 and 3 and is now: "On the capability of the future ALTIUS UV-VIS-NIR limb sounder to constrain modelled stratospheric ozone"
- 2. We also realized that the ALTIUS simulated profiles have inherited some noise from NR since the EnKF procedure adds small perturbations to each model state at each model time step. This would explain some of the larger variability and lower correlation in the comparison between (AR,NR) and (CR,NR) which was pointed out by referee 1 & 2. This is now discussed in the paper with the help of the error scaling profiles calculated by the Desroziers method for NR and AR which are shown. This is something which has not been anticipated before the first submission of the manuscript.
- 3. Two additional experiments have been added in order to evaluate the impact of the low sampling of ALTIUS during the night, a questions raised by referee 1. They consider the assimilation of MLS all data and MLS daytime data with a system configured as for the assimilation run.

**Responses**

The topic of the paper is to show how assimilation of measurements from the upcoming ALTIUS instrument will improve our understanding of the atmospheric state. The paper is lucidly written and well referenced. The title is in my mind too general. *The title has been revised, see the introduction above.*

I have only one comment about the content of the paper:

A fundamental question (in my mind): What is the real purpose of this paper and study? I think that you already have a confirmation that your instrument will fly and that must be congratulated! If results from this study were bad (ALTIUS measurements do not offer any additional constrain), I think you would like to hide your results in your drawer. But I guess that you can be happy with the results.

It is correct to say that ALTIUS is currently under development, and that this study comes after the mission has been accepted for development by ESA. In particular, the space segment is in phase C (detailed design), while the ground segment will terminate its phase B by the end of 2021. However, there remains a number of challenges faced by the industrial consortium. Not the least of them is the relative complexity of the measurement plan: the baseline scenario is a repetition of the primary sequence "bright limb  $\rightarrow$  solar occultation  $\rightarrow$ stellar occultations  $\rightarrow$  solar occultation  $\rightarrow$  bright limb". In particular, the stellar occultations are the most complex to implement, and their implementation is yet to be fully confirmed. One of the key conclusions of this study is that, even small in number, the occultations performed by ALTIUS do make sense from the point of view of data assimilation. This conclusion reinforce the importance of the stellar occultations in the mission planning.

In the introduction, we changed "Second, we also want to measure the added value of the different ALTIUS modes of observation (solar, stellar, planetary and lunar occultations, and bright limb measurements) in particular during the polar night where bright limb observations will not be available" by "Second, we also want to measure the added value of the different ALTIUS modes of observation (solar, stellar, planetary and lunar occultations, and bright limb measurements) in particular for the stellar occultations which are the most complex to implement in the mission scenario".

But this conclusion does not lead to any further developments because a real ALTIUS data simulator (if such an entity exists) has not been used in getting the results.

A study like this would be interesting if data from a realistic data simulator were used or when real data from flying instrument will start flowing in. Results could affect the ALTIUS mission planning and provide help in determining possible biases between various ALTIUS measurement modes and biases with respect to results from other instruments. One of the novelty of ALTIUS is that it pushes the geographical coverage limits of UV-VIS-NIR limb sounders to their maximum by combining bright limb, solar, and stellar occultations. The results contained in this study stem for the combination of all possible observation modes. This study could be used by other teams willing to propose new UV-VIS-NIR limb sounders: our results can help balancing the importance of different observation geometries, and help justifying the appropriateness of a new limb sounder. The assumptions made on the instrument performance, i.e. applying a strict compliance to the ALTIUS signal-to-noise ratio requirements, do not make our conclusions only ALTIUS-relevant. As our requirements are not very different than those considered for other missions (see Bovensmann et al., 1999; Rault and Xu, 2011; Bourassa et al., 2012; these references are now cited in Sect. 4.3.2), one cannot expect dramatically different results with another sensor. Hence we believe that our conclusions are realistic, and reach beyond the scope of ALTIUS.

Bovensmann, H., Burrows, J. P., Buchwitz, M., Frerick, J., Noël, S., Rozanov, V. V., ... Goede, A. P. H. (1999). SCIAMACHY: Mission Objectives and Measurement Modes. Journal of the Atmospheric Sciences, 56(2), 127–150.

Didier F. Rault, Philippe Q. Xu, "Expected data quality from the upcoming OMPS/LP mission," Proc. SPIE 8177, Remote Sensing of Clouds and the Atmosphere XVI, 817709 (26 October 2011); https://doi.org/10.1117/12.897848

Bourassa, A. E., McLinden, C. A., Bathgate, A. F., Elash, B. J., and Degenstein, D. A. (2012), Precision estimate for Odin-OSIRIS limb scatter retrievals, J. Geophys. Res., 117, D04303, doi:10.1029/2011JD016976.